# Inverse-Weighted Survival Games

**Xintian Han** *
NYU
xintian.han@nyu.edu

**Mark Goldstein** *
NYU
goldstein@nyu.edu

**Aahlad Puli**
NYU
aahlad@nyu.edu

**Thomas Wies**
NYU
wies@cs.nyu.edu

**Adler J. Perotte**
Columbia University
adler.perotte@columbia.edu

**Rajesh Ranganath**
NYU
rajeshr@cims.nyu.edu

## Abstract

Deep models trained through maximum likelihood have achieved state-of-the-art results for survival analysis. Despite this training scheme, practitioners evaluate models under other criteria, such as binary classification losses at a chosen set of time horizons, e.g. Brier score (BS) and Bernoulli log likelihood (BLL). Models trained with maximum likelihood may have poor BS or BLL since maximum likelihood does not directly optimize these criteria. Directly optimizing criteria like BS requires inverse-weighting by the censoring distribution. However, estimating the censoring model under these metrics requires inverse-weighting by the failure distribution. The objective for each model requires the other, but neither are known. To resolve this dilemma, we introduce *Inverse-Weighted Survival Games*. In these games, objectives for each model are built from re-weighted estimates featuring the other model, where the latter is held fixed during training. When the loss is proper, we show that the games always have the true failure and censoring distributions as a stationary point. This means models in the game do not leave the correct distributions once reached. We construct one case where this stationary point is unique. We show that these games optimize BS on simulations and then apply these principles on real world cancer and critically-ill patient data.

## 1 Introduction

Survival analysis is the modeling of time-to-event distributions and is widely used in healthcare to predict time from diagnosis to death, risk of disease recurrence, and changes in level of care. In survival data, events, known as *failures*, are often right-censored, i.e., only a lower bound on the time is observed, for instance, when a patient leaves a study before failing. Under certain assumptions, maximum likelihood estimators are consistent for survival modeling [Kalbfleisch and Prentice, 2002].

Recently, deep survival models have obtained state-of-the-art results [Ranganath et al., 2016, Alaa and van der Schaar, 2017, Katzman et al., 2018, Kvamme et al., 2019, Zhong and Tibshirani, 2019]. Common among these are discrete-time models [Yu et al., 2011, Lee et al., 2018, Fotso, 2018, Lee et al., 2019, Ren et al., 2019, Kvamme and Borgan, 2019b, Kamran and Wiens, 2021, Goldstein et al., 2020, Sloma et al., 2021] even when data are continuous because they can borrow classification architectures and flexibly approximate continuous densities [Miscouridou et al., 2018].

Though training is often based on maximum likelihood, criteria such as Brier score (BS) and Bernoulli log likelihood (BLL) have been used to evaluate survival models [Haider et al., 2020]. The BS and BLL are classification losses adapted for survival by treating the model as a binary classifier at various time

---

*Equal Contribution.

35th Conference on Neural Information Processing Systems (NeurIPS 2021).

horizons (*will the event occur before or after 5 years?*) [Kvamme and Borgan, 2019b, Lee et al., 2019, Steingrimsson and Morrison, 2020]. BS can also be motivated by calibration (section 3) which is valuable because survival probabilities are used to communicate risk [Sullivan et al., 2004]. However BS and BLL are challenging to estimate because they require inverse probability of censor-weighting (IPCW), which depends on the true censoring distribution [Van der Laan et al., 2003].

Though consistent, due to finite data, maximum likelihood may lead to models with poor BS and BLL. But directly optimizing these criteria is challenging because IPCW estimation requires solving an additional survival modeling problem to estimate the unknown censoring distribution. This poses a re-weighting dilemma: each model is required for training the other under these criteria but neither are known.

To resolve the dilemma, we introduce *Inverse-Weighted Survival Games* for training with respect to criteria such as BS and BLL. We pose survival analysis as a game with the failure and censoring models as players. Each model's loss is built from IPCW estimates featuring the other model. Inspired by game theory [Neumann and Morgenstern, 2007, Letcher et al., 2019], we ask: should the censoring model's re-weighting role in the failure objective be considered part of the censoring objective? We find the answer to be no. In each step of training, each model follows gradients of its loss with the other model held fixed to compute weights.

When the loss is *proper* (e.g. BS, BLL) [Gneiting and Raftery, 2007], we show that games have the true failure and censoring distributions as a stationary point. This means the models in the game do not leave the correct distributions once reached. We then describe one case where this stationary point is unique. Finally, we show that inverse-weighted game training achieves better BS and BLL than maximum likelihood methods on simulations and real world cancer and ill-patient data.[2]

## 2 Notation and background on IPCW

**Notation.** Let $T$ be a failure time with CDF $F(t) = P(T \le t)$, density $f$, survival function $\overline{F} = 1 - F$, and model $F_{\theta_T}$. Let $C$ be a censoring time with CDF $G$, density $g$, $\overline{G} = 1 - G$, and model $G_{\theta_C}$. This means $\overline{G}(t) = P(C > t)$. Let $\overline{G}(t^-)$ denote $P(C \ge t)$. We observe features $X$, time $U = \min(T, C)$ and $\Delta = \mathbb{1}[T \le C]$. For discrete models over $K$ times, let $\theta_{Tt} = P_\theta(T = t)$ and $\theta_{Ct} = P_\theta(C = t)$.

**Models.** We focus on deep discrete models like those studied in Lee et al. [2018], Kvamme and Borgan [2019b]. The model maps inputs $X$ to a categorical distribution over times. When the observations are continuous, a discretization scheme is necessary. Following Kvamme and Borgan [2019b], Goldstein et al. [2020], we set bins to correspond to quantiles of observed times. We represent all times by the lower boundary of their respective interval.

**Assumptions.** We assume i.i.d. data and random censoring: $T \perp\!\!\!\perp C \mid X$ [Kalbfleisch and Prentice, 2002]. We also require the censoring positivity assumption [Gerds et al., 2013]. Let $f = dF$. Then:

$$\exists \epsilon \quad \text{s.t.} \quad \forall x \, \forall t \in \{t \le t_{\max} \mid f(t|x) > 0\}, \quad \overline{G}(t^-|x) \ge \epsilon > 0, \tag{1}$$

i.e. it is possible that censoring events occur late-enough for us to observe failures up until a maximum time $t_{\max}$. Truncating at a maximum time is necessary in practice for continuous distributions because datasets may have no samples in the tails, leading to practical positivity violations [Gerds et al., 2013]. This truncation happens implicitly for categorical models by choosing the last bin.

In this work, we model the censoring distribution. This task is dual to the original survival problem: the roles of censoring and failure times are reversed. Therefore, to observe censoring events properly, we also require a version of eq. (1) to hold with the roles of $F$ and $G$ reversed (appendix A).

**IPCW estimators.** Inverse probability of censor-weighting (IPCW) is a method for estimation under censoring [Van der Laan et al., 2003, Bang and Robins, 2005]. Consider the marginal mean $\mathbb{E}[T]$. IPCW reformulates such expectations in terms of observed data. Using IPCW, we can show that:

$$\mathbb{E}[T] = \mathop{\mathbb{E}}_{X} \mathop{\mathbb{E}}_{T|X} \left[ \frac{\mathbb{E}[\mathbb{1}[T \le C|X]]}{\mathbb{E}[\mathbb{1}[T \le C|X]]} T \right] = \mathop{\mathbb{E}}_{X} \mathop{\mathbb{E}}_{T|X} \mathop{\mathbb{E}}_{C|X} \left[ \frac{\mathbb{1}[T \le C]}{\overline{G}(T^-|X)} T \right] = \mathop{\mathbb{E}}_{T,C,X} \left[ \frac{\Delta U}{\overline{G}(U^-|X)} \right]$$

---

[2]Code is available at https://github.com/rajesh-lab/Inverse-Weighted-Survival-Games

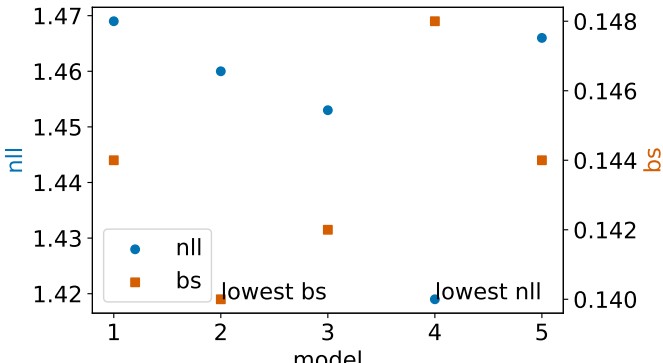

**Figure 1:** Test NLL and BS for 5 different models, each trained with a different learning rate

We derive this fully in appendix B. The second equality holds because $\Delta = 1 \implies U = T$ and means we can identify $\mathbb{E}[T|X]$ provided that we know $G$ and that random censoring and positivity hold.

## 3 Time-dependent survival evaluations

Brier score (BS) [Brier and Allen, 1951] is *proper* for classification, meaning that it has a minimum at the true data distribution [Gneiting and Raftery, 2007]. The BS is often adapted for survival evaluations [Lee et al., 2019, Kvamme et al., 2019, Haider et al., 2020]. For time $t$, it computes differences between the CDF and true event status at $t$, turning survival analysis into a classification problem at a given time horizon:

$$\text{BS}(t; \theta) = \mathbb{E}\left[\left(F_{\theta_T}(t \mid X) - \mathbb{1}\left[T \leq t\right]\right)^2\right] \tag{2}$$

BS is often used as a proxy for marginal calibration error [Kumar et al., 2018, Lee et al., 2019], which measures differences between CDF levels $\alpha \in [0, 1]$ and observed proportions of datapoints with $F_\theta(T|X) < \alpha$ [Demler et al., 2015]. This usage of BS stems from its decomposition into calibration plus a refinement (discriminative) term [DeGroot and Fienberg, 1983].

Unfortunately one cannot compute BS unmodified since $\mathbb{1}\left[T \leq t\right]$ is unobserved for a point censored before $t$. IPCW BS [Graf et al., 1999, Gerds and Schumacher, 2006] estimates BS$(t)$ under censoring:

$$\text{BS}(t; \theta) = \mathbb{E}\left[\frac{\overline{F}_{\theta_T}(t \mid X)^2 \Delta \mathbb{1}\left[U \leq t\right]}{\overline{G}(U^- \mid X)} + \frac{F_{\theta_T}(t \mid X)^2 \mathbb{1}\left[U > t\right]}{\overline{G}(t \mid X)}\right]. \tag{3}$$

eq. (3) is equivalent to eq. (2) (appendix C). Negative Bernoulli log likelihood (BLL) is similar, but with log loss (appendix D). BS and BLL are proper for classification at each time $t$, so their sum or integral over $t$ is still proper (appendix H).

**Proper objectives differ.** Though negative log likelihood (NLL), BS and BLL all have the same true distribution at optimum with infinite data, they may yield significantly different solutions in practice. For example, NLL-trained models may not achieve good BS [Kvamme and Borgan, 2019a]. In fig. 1, we show test set NLL and BS for 5 models trained with NLL at different learning rates on Gamma-simulated data (described in section 5.1). NLL does not align with BS: models that have low NLL may not have low BS. Model 4 has the lowest NLL but not the lowest BS. When a practitioner requires good performance under BS or BLL, they should optimize directly for those metrics.

**Re-weighting dilemma.** Censoring introduces challenges because we must use IPCW to estimate BS and BLL. Crucially, the $G$ in eq. (3) is the true censoring distribution rather than a model, but during training, we only have access to models. This poses a dilemma: can the models be used in re-weighting estimates during training to successfully optimize these criteria under censoring?

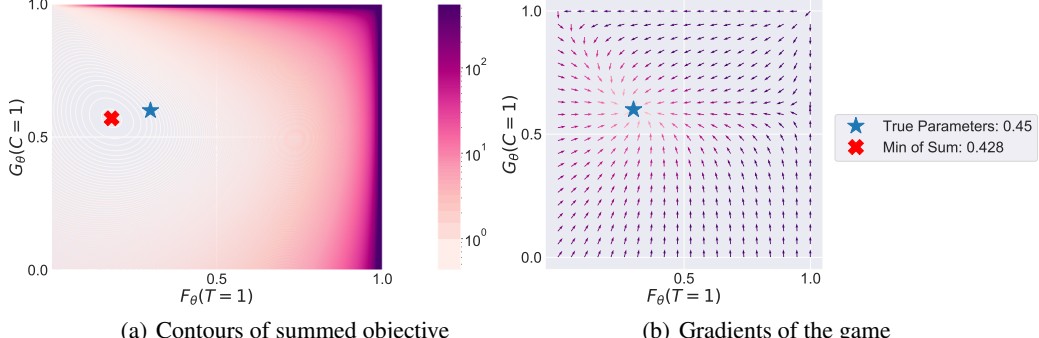

(a) Contours of summed objective      (b) Gradients of the game

**Figure 2:** Figure 2(a): the sum of $F$ and $G$'s IPCW BS(1) scores is an improper joint objective for the failure and censoring models. Figure 2(b): in contrast, as shown in the gradient field, the one timestep game has a unique stationary point at the true data generating distributions.

## 4   Inverse-Weighted Survival Games

A reasonable attempt to solve the dilemma is to jointly optimize the sum of $F_\theta$ and $G_\theta$'s losses where each model re-weights the other's loss. The expectation is that both models will improve over training and yield reliable IPCW estimates for each other. Concretely, consider this for eq. (3) plus the same objective with the roles of $F_\theta$ and $G_\theta$ reversed. Unfortunately, there exist solutions to this optimization problem with smaller loss than for the distributions from which the data was generated, making this summed objective improper for the pair of distributions. In fig. 2, we plot this for IPCW BS$(t = 1)$ for models over two timesteps[3] as a function of each model's single parameter.

To address this phenomenon, we introduce *Inverse-Weighted Survival Games*. In these games, a *failure player* and *censor player* simultaneously minimize their own loss function. The failure and censoring model are featured in both loss functions and playing the game results in a trained failure and censoring model. We show in experiments that these games produce models with good BS, BLL, and concordance relative to those trained with maximum likelihood.

For simplicity, we present the games for marginal categorical models. The analysis can be extended to conditional parameterizations with the usual caveats shared by maximum likelihood. Our experiments explore the conditional setting.

### 4.1   Basic definition of game

We follow the setup in Letcher et al. [2019]. A differentiable $n$-player game consists of $n$ players each with loss $\ell_i$ and parameter (or state) $\theta_i$. Player $i$ controls *only* $\theta_i$ and aims to minimize $\ell_i$. However, each $\ell_i$ is a function of the whole state $\theta = (\theta_i, \theta_{-i})$[4]. The *simultaneous gradient* is the gradient of the losses w.r.t. each players' respective parameters:

$$\xi(\theta) = [\nabla_{\theta_1}\ell_1, \ldots, , \nabla_{\theta_n}\ell_n]$$

The *dynamics* of the game refers to following $-\xi$. The solution concepts in games are equilibria (the game analog of optima) and stationary points. One necessary condition for equilibria is finding a *stationary point* $\theta^\star$ such that $\xi(\theta^\star) = 0$. The simplest algorithm follows the dynamics to find stationary points and is called simultaneous gradient descent. With learning rate $\eta$,

$$\theta \leftarrow \theta - \eta\xi(\theta).$$

This can be interpreted as each player taking their best possible move at each instant.

---

[3]BS$(t = 1)$ is proper for distributions with support over two timesteps because BS$(t = K)$ for a model with support over $K$ timesteps is always 0, so the summed BS equals BS(1).

[4]For two players, when $\ell_1 = -\ell_2$ the game is called *zero-sum* and can be written as a minimax game, sometimes referred to as *adversarial* (e.g. as in Generative Adversarial Networks (GANs) [Goodfellow et al., 2014]). Games with $\ell_1 \neq -\ell_2$ are called non-zero-sum.

## 4.2 Constructing survival games

We specify an Inverse-Weighted Survival Game as follows. First, choose a loss $L$ used to construct the losses for each player. Next, derive the IPCW form $L_I$ that can be used to compute $L$ under censoring: for true failure and censoring distributions $F^\star$ and $G^\star$, the losses $L$ and $L_I$ are related through $L_I(F_{\theta_T}; G^\star) = L(F_{\theta_T})$ and $L_I(G_{\theta_C}; F^\star) = L(G_{\theta_C})$. The loss functions for the two players are defined as:

$$\ell_F(\theta) \triangleq L_I(F_{\theta_T}; G_{\theta_C}), \quad \ell_G(\theta) \triangleq L_I(G_{\theta_C}; F_{\theta_T}) \tag{4}$$

Compared to eq. (3), we have replaced the true re-weighting distributions with models. Finally, the failure player and censor player minimize their respective loss functions w.r.t. only their own parameters:

$$\texttt{failure player:} \min_{\theta_T} \ell_F, \quad \texttt{censor player:} \min_{\theta_C} \ell_G$$

One example of these games is the IPCW $\text{BS}(t)$ game, derived in appendix C. With $\overline{\Delta} = 1 - \Delta$,

$$
\begin{aligned}
\ell_F^t(\theta) &= \mathbb{E}\left[ \frac{\overline{F}_{\theta_T}(t)^2 \Delta \mathbb{1}\left[U \le t\right]}{\overline{G}_{\theta_C}(U^-)} + \frac{F_{\theta_T}(t)^2 \mathbb{1}\left[U > t\right]}{\overline{G}_{\theta_C}(t)} \right] \\
\ell_G^t(\theta) &= \mathbb{E}\left[ \frac{\overline{G}_{\theta_C}(t)^2 \overline{\Delta} \mathbb{1}\left[U \le t\right]}{\overline{F}_{\theta_T}(U)} + \frac{G_{\theta_C}(t)^2 \mathbb{1}\left[U > t\right]}{\overline{F}_{\theta_T}(t)} \right].
\end{aligned}
\tag{5}
$$

In section 4.3, we show that this formulation (fig. 2(b)) has formal advantages over the optimization in fig. 2(a) for particular choices of $L$.

**Multiple Timesteps.** The example is specified for a given $t$, but the games can be designed for multiple timesteps. We use BS for a $K$ timestep model to demonstrate. $\text{BS}(K)$ is 0 for any model: the left terms contain $\overline{F}_{\theta_T}$ and $\overline{G}_{\theta_C}$, which are are both 0 when evaluated at $K$; in the right terms, $\mathbb{1}\left[U > K\right]$ is always 0. One option is to define summed games with:

$$\ell_F = \sum_{t=1}^{K-1} \ell_F^t, \quad \ell_G = \sum_{t=1}^{K-1} \ell_G^t$$

The summed game is shown in algorithm 1. Alternatively, instead of the sum, it is possible to find solutions for all timesteps with respect to one pair of models $(F_\theta, G_\theta)$. For $K$-1 timesteps this can be formalized as a $2(K\text{-}1)$-player game: there is a failure player and censor player for the loss at each $t$:

$$t^{th}\texttt{-failure player:} \min_{\theta_{Tt}} \ell_F^t, \quad t^{th}\texttt{-censor player:} \min_{\theta_{Ct}} \ell_G^t$$

We study theory that applies to both approaches in section 4.3, namely that the true failure and censoring distribution are stationary points in both types of games. We prove additional properties about uniqueness of the stationary point for a special case of the multi-player game in section 4.4. Summed games are more stable to optimize in practice because they optimize objectives at all time steps w.r.t all parameters, while multiplayer games can only improve each time step's loss w.r.t. one parameter. We study the summed games empirically in section 5.

---

**Algorithm 1** Following Gradients in Summed Games
***
    **Input:** Choice of losses $\ell_F, \ell_G$, learning rate $\gamma$
    **Initialize:** $T$ model parameters $\theta_T$ and $C$ model parameters $\theta_C$ randomly
    **repeat**
        Set $g_T = 0$ and $g_C = 0$
        **for** $t = 1$ **to** $K - 1$ **do**
            $g_T = g_T + d\ell_F^t/d\theta_T$
            $g_C = g_C + d\ell_G^t/d\theta_C$
        **end for**
        $\theta_T \leftarrow \theta_T - \gamma g_T$ and $\theta_C \leftarrow \theta_C - \gamma g_C$
    **until** convergence
    **Output:** $\theta_T, \theta_C$

---

### 4.3  IPCW games have a stationary point at data distributions

Among a game's stationary points should be the true failure and censoring distributions.

**Proposition 1.** Assume $\exists \theta_T^\star \in \Theta_T, \exists \theta_C^\star \in \Theta_C$ such that $F^\star = F_{\theta_T^\star}$ and $G^\star = G_{\theta_C^\star}$. Assume the game losses $\ell_F, \ell_G$ are based on proper losses $L$ and that the games are only computed at times for which positivity holds. Then $(\theta_T^\star, \theta_C^\star)$ is a stationary point of the game eq. (4).

The proof is in appendix I. The result holds for summed and multi-player games using BS, BLL, or other proper scoring rules such as AUC.[5] When games are built from such objectives, the set of solutions includes $(\theta_T^\star, \theta_C^\star)$ and models do not leave this correct solution when reached. Under the stated assumptions, this result holds for discrete and continuous distributions. However, as mentioned in section 2, in practice a truncation time must be picked to ensure the assumptions are met for continuous distributions.

### 4.4  Uniqueness for Discrete Brier Games

We provide a stronger result for the BS game in eq. (5) when solving all timesteps with multi-player games: its only stationary point is located at the true failure and censoring distributions.

**Proposition 2.** For discrete models over $K$ timesteps, assuming that $\theta_{Tt}^\star > 0$ and $\theta_{Ct}^\star > 0$, the solution $(\theta_T^\star, \theta_C^\star)$ is the only stationary point for the multi-player BS game shown in algorithm 2 for times $t \in \{1, \ldots, K-1\}$

The proof is in appendix J. To illustrate this, fig. 2(b) shows that, unlike the minimization in fig. 2(a), the IPCW BS game moves to the correct solution at its unique stationary point.

## 5  Experiments

We run experiments on a simulation with conditionally Gamma times, a semi-simulated survival dataset based on MNIST, several sources of cancer data, and data on critically-ill hospital patients.

**Losses.**  We build categorical models in 3 ways: the standard NLL method (eq. (8)), the IPCW BS game and the negative IPCW BLL game.

**Metrics.**  For these models we report BS (uncensored for simulations and Kaplan-Meier (KM)-weighted for real data), BLL (also uncensored or weighted), concordance which measures the proportion of pairs whose predicted risks are ordered correctly [Harrell Jr et al., 1996], and NLL. We report mean and standard deviation of the metrics over 5 different seeds. In all plots, the middle solid line represents the mean and the shaded band represents the standard deviation.

**Model Description.**  In all experiments except for MNIST, we use a 3-hidden-layer ReLU network that outputs 20 categorical bins (more bin choices in appendix G.1). For MNIST we first use a small convolutional network and follow with the same fully-connected network.

Model and training details including learning rate and model selection can be found in appendix F.

### 5.1  Simulation Studies

**Data.**  We draw $X \in \mathbb{R}^{32} \sim \mathcal{N}(0, 10I)$ and $T \sim \text{Gamma}(\text{mean} = \mu_t)$ where $\mu_t$ a log-linear function of $X$. The censoring times are also gamma with mean $0.9 * \mu_t$. Both distributions have constant variance $0.05$. It holds that $T \perp\!\!\!\perp C \mid X$. Each random seed draws a new dataset.

**Results.**  fig. 3 demonstrates that the games optimize the true uncensored BS, and, though more slowly with respect to training size, log-likelihood does too. The games have better test-set performance on all metrics for small training size. All methods converge on similar performance when there is enough data (though *enough* is highly-dependent on dimensionality and model class).

---

[5]Though often reported, the time-dependent concordance($t$) is not proper [Blanche et al., 2019].

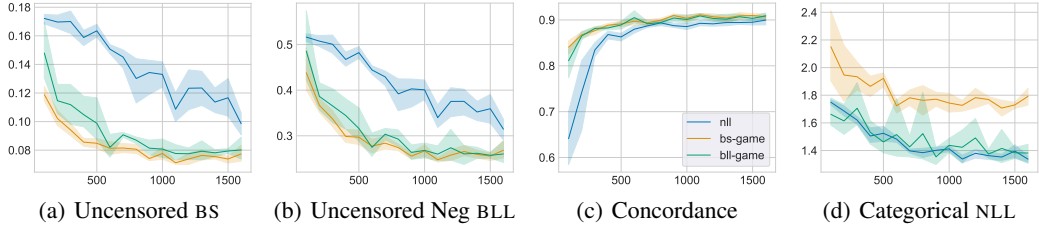

| (a) Uncensored BS | (b) Uncensored Neg BLL | (c) Concordance | (d) Categorical NLL |

**Figure 3:** Test set evaluation metrics (y-axis) on the Gamma simulation versus number of training points (x-axis) for three methods. Each point in the plot represents the evaluation metric value of a fully trained model with that number of training points. Lower is better for all the metrics except for concordance.

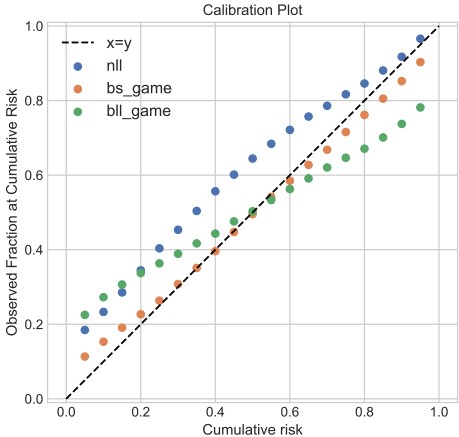

**Figure 4:** Calibration curves [Avati et al., 2019] comparing game-training and NLL-training. .

**Calibration.** We include a qualitative investigation of model calibration on the gamma simulation trained with 2000 datapoints. fig. 4 shows that the BS game achieves near-perfect calibration while the two likelihood-based methods suffer some error. This is expected since likelihood-based methods do not directly optimize calibration while BS does (section 3).

## 5.2 Semi-simulated studies

**Data.** Survival-MNIST [Gensheimer, 2019, Pölsterl, 2019] draws times conditionally on MNIST label $Y$. This means digits define risk groups and $T \perp\!\!\!\perp X \mid Y$. Times within a digit are i.i.d. The model only sees the image pixels $X$ as covariates so it must learn to classify digits (risk groups) to model times. We follow Goldstein et al. [2020] and use Gamma times. $T \sim \text{Gamma}(\text{mean} = 10 * (Y + 1))$. We set the variance constant to $0.05$. Lower digit labels $Y$ yield earlier event times. $C$ is drawn similarly but with mean $9.9 * (Y + 1)$. Each random seed draws a new dataset.

**Results.** This experiment demonstrates that better NLL does not correspond to better performance on BS, BLL, and concordance. Similarly to the previous experiment, fig. 5 shows that game methods attain better uncensored test-set BS and BLL on survival-MNIST than likelihood-based training does. The games likewise attain higher concordance. NLL training performs better at the metric it directly optimizes. This experiment also establishes that it is possible to optimize through deep convolutional models with batch norm and pooling using the game training methods.

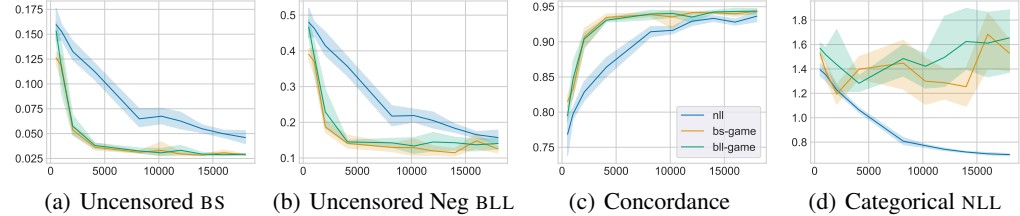

| (a) Uncensored BS | (b) Uncensored Neg BLL | (c) Concordance | (d) Categorical NLL |

**Figure 5:** Test set evaluation metrics (y-axis) on survival-MNIST versus number of training points (x-axis) for three methods. Each point in the plot represents the evaluation metric value of a fully trained model with that number of training points. Lower is better for all the metrics except for concordance.

## 5.3 Real Datasets

**Data.** We use several datasets used in recent papers [Chen, 2020, Kvamme et al., 2019] and available in the python packages DeepSurv [Katzman et al., 2018] and PyCox [Kvamme et al., 2019], and the R Survival [Therneau, 2021]. The datasets are:

- Molecular Taxonomy of Breast Cancer International Consortium (METABRIC) [Curtis et al., 2012]
- Rotterdam Tumor Bank (ROTT) [Foekens et al., 2000] and German Breast Cancer Study Group (GBSG) [Schumacher et al., 1994] combined into one dataset (ROTT. & GBSG)
- Study to Understand Prognoses Preferences Outcomes and Risks of Treatment (SUPPORT) [Knaus et al., 1995] which includes severely ill hospital patients

For more description see appendix F. For real data, there is no known ground truth for the censoring distribution, which means evaluation requires assumptions. Following the experiments in Kvamme et al. [2019], we assume that censoring is marginal estimate with KM to evaluate models.[6]

**Results.** On METABRIC, games attain lower (better) KM-weighted BS and BLL than NLL-training when the number of datapoints is small, and have better concordance and NLL though they do not directly optimize them. As data size increases, all methods converge to similar performance. On ROTT. & GBSG, the trend is similar: games optimize the BS and BLL more rapidly as a function of training set size than NLL-training does. Again, all methods converge to similar performance in all metrics when the number of datapoints is large enough. All methods perform similarly on SUPPORT.

**Caveats.** First, though popular, these survival datasets are low-dimensional, so any of the objectives can perform well on the metrics with just several hundred points. We see that this is distinct from MNIST, where thousands of points were required to improve performance. Second, though possible, it may not be true that censoring is marginal on these datasets, which would mean that the BS and BLL results only have their interpretation conditional on a particular set of covariates. Our method is also correct when the censoring is marginal though. Lastly, no method is stable for all metrics, for all training sizes, on all seeds for all datasets.

## 6 Related Work

**Nuisance parameters.** Under non-informative censoring, the censoring distribution is unrelated to the failure distribution, but estimating it can help improve learning the failure distribution; here, the censoring distribution is a *nuisance parameter*. Existing causal estimation methods propose two-stage procedures where the first stage estimates the nuisance-parameter (e.g. propensity score) and the second stage uses the learned nuisance-parameter as-is to define an estimator or loss function for the target parameter (causal effect). [Van Der Laan and Rubin, 2006, Van der Laan and Rose, 2011, Chernozhukov et al., 2018, Foster and Syrgkanis, 2019]. In this work, we instead show that

---

[6]This is also the route taken in the R packages Survival [Therneau, 2021], PEC [Mogensen et al., 2012], and riskRegression [Gerds et al., 2020].

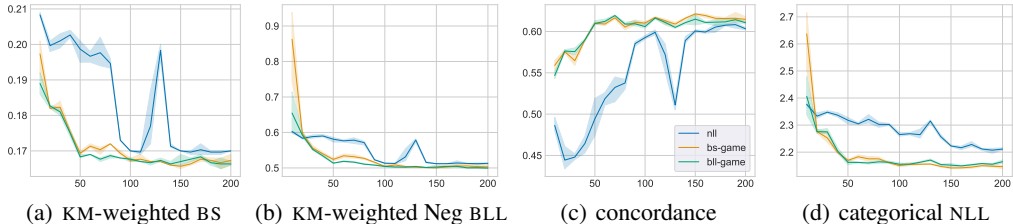

(a) KM-weighted BS  (b) KM-weighted Neg BLL  (c) concordance  (d) categorical NLL

**Figure 6:** Test set evaluation metrics (y-axis) on METABRIC versus number of training points (x-axis) for three methods. Each point in the plot represents the evaluation metric value of a fully trained model with that number of training points. Lower is better for all the metrics except for concordance.

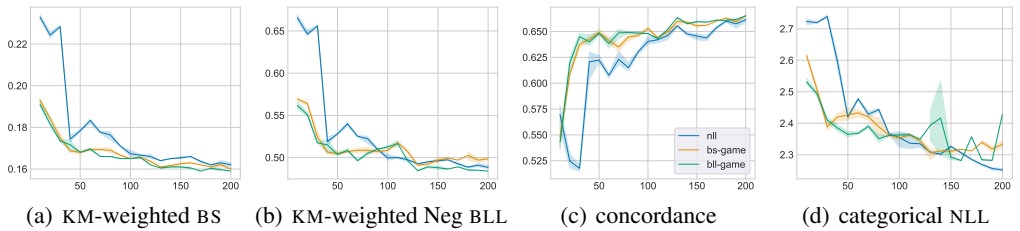

(a) KM-weighted BS  (b) KM-weighted Neg BLL  (c) concordance  (d) categorical NLL

**Figure 7:** Test set evaluation metrics (y-axis) on ROTT. & GBSG versus number of training points (x-axis) for three methods. Each point in the plot represents the evaluation metric value of a fully trained model with that number of training points. Lower is better for all the metrics except for concordance.

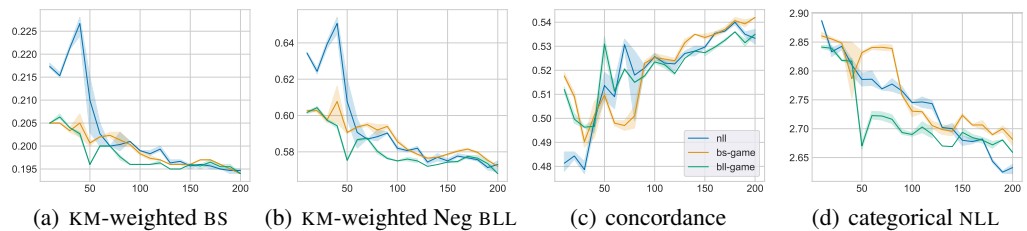

(a) KM-weighted BS  (b) KM-weighted Neg BLL  (c) concordance  (d) categorical NLL

**Figure 8:** Test set evaluation metrics (y-axis) on SUPPORT versus number of training points (x-axis) for three methods. Each point in the plot represents the evaluation metric value of a fully trained model with that number of training points. Lower is better for all the metrics except for concordance.

estimating the target (failure model objective or failure model itself) can benefit from a *coupled* estimation procedure where the nuisance parameter (censoring model) is also trained simultaneously. The failure model needs the censoring distribution to compute BS but censoring estimation needs the failure model, and despite this circular dependence, we characterize a case where the game training leads to the true data generating distributions.

**Double Robust Censoring Unbiased Transformations.**   For functions $h$, Rubin and van der Laan [2007] estimate conditional mean $\mathbb{E}[h(T, X)|X]$ under censoring using a double-robust estimator: given estimates of the conditional failure and censoring cumulative distribution functions (CDFs) $\hat{F}(t|X)$ and $\hat{G}(c|X)$, the estimator of $\mathbb{E}[h(T, X)|X]$ is unbiased when either nuisance CDF is correct. However, here we are concerned with estimating a quantity to be used as a loss for learning $\hat{F}$. We therefore presumably do not already have an estimate of $\hat{F}$ to be used in a doubly-robust estimator.

**Censoring Unbiased Losses for Deep Learning.**   Steingrimsson and Morrison [2020] build failure model loss functions based on the estimators from Rubin and van der Laan [2007]. Their BS loss extends IPCW BS estimation to the doubly-robust case and to our knowledge is the first instance of

IPCW-based estimation procedures being used in a general purpose way to define loss functions for deep survival analysis.

However, their censoring distribution is estimated once before training and held fixed rather than incorporated into a joint training procedure as in the games in this work. The fixed censoring estimate is implemented by KM, which assumes a marginal censoring distribution. Making the marginal assumption when censoring is truly conditional should not yield a performant model under the BS criteria since the training objective does not directly estimate or optimize the true BS that would be measured under no censoring. When marginal censoring does hold, KM estimation, which is non-parametric, may be a simpler and stable choice versus the game, depending on sample size, data variance, and conditional parameterization assumptions. But since it is in general unknown if censoring is marginal, we use conditional models which are also correct under marginal censoring.

## 7 Discussion

In this work, we propose a new training method for survival models under censored data. We argue that on finite data, it is important to close the gap between training methodology and the desired evaluation criteria. We showed in the experiments that better NLL does not correspond to better performance on BS, BLL, and concordance, all evaluations of interest in survival analysis.

The main trend in our experimental results was that data size matters: smaller meant the game methods performed better than NLL and enough data meant that they perform similarly, which is expected since all objectives are proper. However *enough* data is hard to define: it depends on dimensionality and on the data generating distribution and model class. It is a great direction to build more precise understanding on how objectives behave differently even when they have the same optimum on infinite data: though likelihood is known to be asymptotically efficient for survival analysis, more analysis is necessary for comparing likelihood and Brier score's trade-offs on small sample sizes.

In the experiments, we focus on categorical models. On the other hand, proposition 1 applies to continuous distributions as well, provided that positivity can be satisfied. However, this is rare in practice because most survival data has a final follow-up time, and even before this time there may be very few samples with late times [Gerds et al., 2013]. For this reason, working with continuous distributions requires picking a truncation time and playing games only up to that time.

Evaluation on real data under censoring requires assumptions. It is important to further consider how to better assess test-set performance on metrics such as BS, BLL, and concordance. Because concordance is not proper [Blanche et al., 2019], we do not build objectives from it here, but it too is not invariant to censoring. Regarding games, we showed properties about stationary points. More analysis is necessary to describe important convergence properties of optimizing these games.

**Social Impact.** Survival models are deployed in hospital settings and have high impact on public health. In this work, we saw benefits of a new training approach for these models, but no training method is a panacea. Practitioners of survival analysis must take great care to consider various training and validation approaches, as well as consider possible test distribution shifts, prior to deployment.

## Acknowledgments and Disclosure of Funding

This work was supported by:

- NIH/NHLBI Award R01HL148248
- NSF Award 1922658 NRT-HDR: FUTURE Foundations, Translation, and Responsibility for Data Science.
- NSF Award 1514422 TWC: Medium: Scaling proof-based verifiable computation
- NSF Award 1815633 SHF

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
