# A  Notation, Assumptions, and Likelihoods in More Detail

## A.1  Notation

Let $T$ be a failure time with CDF $F$. $T$'s *survival function* is defined by $\overline{F} = 1 - F$. We denote failure models by $F_{\theta_T}$. Let $C$ be a censoring time with CDF $G$, survival function $\overline{G}$, and model $G_{\theta_C}$. Under right-censoring, define $U = \min(T, C)$, $\Delta = \mathbb{1}\left[T \leq C\right]$ and we observe $(X_i, U_i, \Delta_i)$. We use $\overline{G}(t^-)$ to denote $P(C \geq t)$.

## A.2  Assumptions

We assume i.i.d. data and random censoring: $T \perp\!\!\!\perp C \mid X$ [Kalbfleisch and Prentice, 2002]. Derivations in this work also require the censoring positivity assumption [Gerds et al., 2013]. Let $f = dF$ (a failure density) and $g = dG$ (a censoring density). Then we assume

$$\exists \epsilon \quad \text{s.t.} \quad \forall x \, \forall t \in \{t \leq t_{\max} \mid f(t|x) > 0\}, \quad \overline{G}(t^-|x) \geq \epsilon > 0, \tag{6}$$

for some truncation time $t_{\max}$. Truncating at a maximum time is necessary in practice for continuous distributions because datasets may have no samples in the tails, leading to practical positivity violations [Gerds et al., 2013]. This truncation happens implicitly for categorical models by choosing the bins.

To observe censoring events properly, we also require a version of eq. (1) to hold with the roles of $F$ and $G$ reversed:

$$\exists \epsilon \quad \text{s.t.} \quad \forall x \, \forall t \in \{t \leq t_{\max} \mid g(t|x) > 0\}, \quad \overline{F}(t|x) \geq \epsilon > 0. \tag{7}$$

$t_{\max}$ should be chosen so that these two conditions hold.

## A.3  Likelihoods

As mentioned, we assume data are i.i.d. and censoring is random $T \perp\!\!\!\perp C \mid X$. Under these assumptions, the likelihood, by definition [Andersen et al., 2012], is:

$$L(\theta_T, \theta_C) = \prod_i \left[ f_{\theta_T}(U_i) \overline{G}_{\theta_C}(U_i^-) \right]^{\Delta_i} \left[ g_{\theta_C}(U_i) \overline{F}_{\theta_T}(U_i) \right]^{1 - \Delta_i}, \tag{8}$$

When a failure is observed, $\Delta_i = \mathbb{1}\left[T_i \leq C_i\right] = 1$ so we compute the failure density or mass $f$ at the observed time $U_i = T_i$. In this case, the only thing we know about the censoring time is $C_i \geq T_i = U_i$. We therefore compute $P(C_i \geq T_i) = P(C_i \geq U_i) = 1 - G_{\theta_C}(U_i^-) = \overline{G}_{\theta_C}(U_i^-)$. Likewise, when a censoring time is observed, $\Delta_i = 0$ so we compute the censoring density or mass $g$ at the observed censoring time $U_i = C_i$. In this case, the only thing we know about the failure time is that $T_i > C_i$. We therefore compute $P(T_i > C_i) = P(T_i > U_i) = 1 - F(U_i) = \overline{F}(U_i)$.

Under the additional assumption of non-informativeness -that $F$ and $G$ don't share parameters and therefore $\theta_T, \theta_C$ are distinct- the $g/G$ terms are constant wrt $\theta_T$ and the $f/F$ terms are constant wrt $\theta_C$. In this case, when one is modeling failures, they can use the partial failure likelihood:

$$L(\theta_T)^{\text{partial}} = \prod_i \left[ f_{\theta_T}(U_i) \right]^{\Delta_i} \left[ \overline{F}_{\theta_T}(U_i) \right]^{1 - \Delta_i}$$

And when one is modeling censoring they can use the partial censoring likelihood:

$$L(\theta_C)^{\text{partial}} = \prod_i \left[ \overline{G}_{\theta_C}(U_i^-) \right]^{\Delta_i} \left[ g_{\theta_C}(U_i) \right]^{1 - \Delta_i}$$

## A.4  Failure partial likelihood depends on true censoring distribution

We now show that the failure partial likelihood's scale depends on the true sampling distribution of censoring times, even if the censoring model has dropped as a constant in the objective. The expected likelihood is:

$$\mathbb{E}_{\substack{T \sim F_{\theta_T^*}, C \sim G_{\theta_C^*} \\ U = \min(T, C), \Delta = \mathbb{1}[T \leq C]}} \left[ f_{\theta_T}(U)^{\mathbb{1}[\Delta = 1]} \overline{F}_{\theta_T}(U)^{\mathbb{1}[\Delta = 0]} \right]$$

The reason is that $\Delta$ and $U$ depend on T and C (therefore on $F_{\theta_T^*}$ and $G_{\theta_C^*}$). We now constructively show that the failure model's NLL can vary with the true censoring distribution. Let us consider a marginal survival analysis problem (no features) and random censoring. The log NLL is:

$$\underset{F_{\theta_T^*},G_{\theta_C^*}}{\mathbb{E}}\left[\Delta \log f_{\theta_T}(U)\right] + \underset{F_{\theta_T^*},G_{\theta_C^*}}{\mathbb{E}}\left[(1-\Delta)\log \overline{F}_{\theta_T}(U)\right]$$

Now consider an $F_{\theta_T^*}$ whose support starts at time 1 (e.g. uniform over 1,2,3) and $G_{\theta_C^*}$ such that there is probability $\rho$ that $C = 0$ and probability $1 - \rho$ that $C$ take a value above the support of $T$ (e.g. >3). Points are therefore only censored at time 0 or uncensored.

$$\underset{F_{\theta_T^*},G_{\theta_C^*}}{\mathbb{E}}\left[\Delta \log f_{\theta_T^*}(U)\right] + \underset{F_{\theta_T^*},G_{\theta_C^*}}{\mathbb{E}}\left[(1-\Delta)\log \overline{F}_{\theta_T^*}(U)\right]$$

$$= (1-\rho) \underset{F_{\theta_T^*}}{\mathbb{E}}\left[\log f_{\theta_T^*}(T)\right] + \rho \underset{G_{\theta_C^*}}{\mathbb{E}}\left[\log \overline{F}_{\theta_T^*}(C)\right]$$

$$= (1-\rho) \underset{F_{\theta_T^*}}{\mathbb{E}}\left[\log f_{\theta_T^*}(T)\right] + \rho \underset{G_{\theta_C^*}}{\mathbb{E}}\left[\log \overline{F}_{\theta_T^*}(0)\right]$$

$$= (1-\rho) \underset{F_{\theta_T^*}}{\mathbb{E}}\left[\log f_{\theta_T^*}(T)\right] + \rho \underset{G_{\theta_C^*}}{\mathbb{E}}\left[\log 1\right]$$

$$= (1-\rho) \underset{F_{\theta_T^*}}{\mathbb{E}}\left[\log f_{\theta_T^*}(T)\right] + \rho \underset{G_{\theta_C^*}}{\mathbb{E}}\left[0\right]$$

$$= (1-\rho) \underset{F_{\theta_T^*}}{\mathbb{E}}\left[\log f_{\theta_T^*}(T)\right]$$

This quantity depends on $\rho$. This shows that the failure model's NLL depends on the true sampling distribution of censoring times.

## B  IPCW Primer

IPCW is a technique for estimation under censoring [Gerds and Schumacher, 2006]. Consider estimating the marginal mean of $T$ : $\mathbb{E}[T] = \mathbb{E}_X \mathbb{E}_{T|X}[T]$. $T$ is not observed for all datapoints. Instead, we observe $U = \min(T, C)$ and $\Delta = \mathbb{1}[T \leq C]$. IPCW reformulates such expectations in terms of observed data. Using this method, we can show that:

$$\underset{X}{\mathbb{E}}\,\underset{T|X}{\mathbb{E}}[T] = \underset{X}{\mathbb{E}}\,\underset{T|X}{\mathbb{E}}\left[\frac{\mathbb{E}_{C|X}\,\mathbb{1}[T \leq C]}{\mathbb{E}_{C'|X}\,\mathbb{1}[T \leq C']}T\right]$$

$$= \underset{X}{\mathbb{E}}\,\underset{T|X}{\mathbb{E}}\,\underset{C|X}{\mathbb{E}}\left[\frac{\mathbb{1}[T \leq C]}{\mathbb{E}_{C'|X}\,\mathbb{1}[T \leq C']}T\right]$$

$$= \underset{T,C,X}{\mathbb{E}}\left[\frac{\mathbb{1}[T \leq C]}{\mathbb{E}_{C'|X}\,\mathbb{1}[T \leq C']}T\right]$$

$$= \underset{T,C,X}{\mathbb{E}}\left[\frac{\mathbb{1}[T \leq C]}{\mathbb{P}(C' \geq T|X)}T\right]$$

$$= \underset{T,C,X}{\mathbb{E}}\left[\frac{\mathbb{1}[T \leq C]}{\overline{G}(T^-|X)}T\right]$$

$$= \underset{U,\Delta,X}{\mathbb{E}}\left[\frac{\Delta U}{\overline{G}(U^-|X)}\right]$$

We have used $C'$ in the denominator to emphasize that it is not a function of $C$ in the integral over the numerator indicator once that expectation is moved out. We have used random censoring to go from $\mathbb{E}_{T|X}\,\mathbb{E}_{C|X}$ to the joint $\mathbb{E}_{T,C|X}$. The last equality changes from the complete data distribution to the observed distribution and holds because $\Delta = 1 \implies U = T$. This means we can estimate the

expectation, provided that we know $G$ and that random censoring and positivity (eq. (1)) hold. In practice, we must learn the censoring distribution, a challenging task as it is also censored.

Graf et al. [1999] develop the IPCW BS. Gerds and Schumacher [2006] extend it to conditional censoring and Kvamme and Borgan [2019a] specialize to administrative censoring. Gerds et al. [2013], Wolbers et al. [2014] develop the IPCW concordance. IPCW estimators for several forms of area under curve (AUC) have been studied in Hung and Chiang [2010a,b], Blanche et al. [2013, 2019], Uno et al. [2007]. Yadlowsky et al. [2019] derive an IPCW estimator for binary survival calibration.

## C  Deriving IPCW Brier Scores

We derive the IPCW BS introduced by Graf et al. [1999], Gerds and Schumacher [2006]. In the below let F-BS be the F model's BS and let F-BS-CW be its censor-weighted version. The censor-weighted failure BS:

$$\text{F-BS-CW}(t) = \mathop{\mathbb{E}}_{T,C}\left[\frac{(1-F_\theta(t))^2\mathbb{1}\left[T \leq C\right]\mathbb{1}\left[U \leq t\right]}{P_\theta(C' \geq U)} + \frac{F_\theta(t)^2\mathbb{1}\left[U > t\right]}{P_\theta(C' > t)}\right]$$

where $U = \min(T, C)$ and $F_\theta = P_\theta(T \leq \cdot)$, It's relationship to the regular BS is:

$$\text{F-BS}(t) = \mathop{\mathbb{E}}_{T}\left[\left(F_\theta(t) - \mathbb{1}\left[T \leq t\right]\right)^2\right]$$

$$= \mathop{\mathbb{E}}_{T}\left[(1-F_\theta(t))^2\mathbb{1}\left[T \leq t\right] + F_\theta(t)^2\mathbb{1}\left[T > t\right]\right]$$

$$= \mathop{\mathbb{E}}_{T}\left[\frac{\mathbb{E}_C\,\mathbb{1}\left[T \leq C\right]}{\mathbb{E}_{C'}\,\mathbb{1}\left[T \leq C'\right]}(1-F_\theta(t))^2\mathbb{1}\left[T \leq t\right] + \frac{\mathbb{E}_C\,\mathbb{1}\left[C > t\right]}{\mathbb{E}_{C'}\,\mathbb{1}\left[C' > t\right]}F_\theta(t)^2\mathbb{1}\left[T > t\right]\right]$$

$$= \mathop{\mathbb{E}}_{T,C}\left[\frac{(1-F_\theta(t))^2\mathbb{1}\left[T \leq C\right]\mathbb{1}\left[T \leq t\right]}{\mathbb{E}_{C'}\,\mathbb{1}\left[T \leq C'\right]} + \frac{F_\theta(t)^2\mathbb{1}\left[T > t\right]\mathbb{1}\left[C > t\right]}{\mathbb{E}_{C'}\,\mathbb{1}\left[C' > t\right]}\right]$$

$$= \mathop{\mathbb{E}}_{T,C}\left[\frac{(1-F_\theta(t))^2\mathbb{1}\left[T \leq C\right]\mathbb{1}\left[T \leq t\right]}{P_\theta(C' \geq T)} + \frac{F_\theta(t)^2\mathbb{1}\left[T > t\right]\mathbb{1}\left[C > t\right]}{P_\theta(C' > t)}\right]$$

$$= \mathop{\mathbb{E}}_{T,C}\left[\frac{(1-F_\theta(t))^2\mathbb{1}\left[T \leq C\right]\mathbb{1}\left[U \leq t\right]}{P_\theta(C' \geq U)} + \frac{F_\theta(t)^2\mathbb{1}\left[U > t\right]}{P_\theta(C' > t)}\right]$$

$$= \text{F-BS-CW}(t)$$

The expectation comes out due to $T \perp\!\!\!\perp C$. The last line follows from $T \leq C \implies U = T$ (in the left term) and $\mathbb{1}\left[T > t\right]\mathbb{1}\left[C > t\right] = \mathbb{1}\left[U > t\right]$ (in the right term). Define likewise the failure-weighted censor BS

$$\text{G-BS-CW}(t) = \mathop{\mathbb{E}}_{T,C}\left[\frac{(1-G_\theta(t))^2\mathbb{1}\left[C < T\right]\mathbb{1}\left[U \leq t\right]}{P_\theta(T' > U)} + \frac{G_\theta(t)^2\mathbb{1}\left[U > t\right]}{P_\theta(T' > t)}\right]$$

where $G_\theta = P_\theta(C \leq \cdot)$. The relationship to the censoring distribution's BS is:

$$\text{G-BS}(t) = \mathop{\mathbb{E}}_{C}\left[\left(G_\theta(t) - \mathbb{1}\left[C \leq t\right]\right)^2\right]$$

$$= \mathop{\mathbb{E}}_{C}\left[(1-G_\theta(t))^2\mathbb{1}\left[C \leq t\right] + G_\theta(t)^2\mathbb{1}\left[C > t\right]\right]$$

$$= \mathop{\mathbb{E}}_{C}\left[\frac{\mathbb{E}_T\,\mathbb{1}\left[C < T\right]}{\mathbb{E}_{T'}\,\mathbb{1}\left[C < T'\right]}(1-G_\theta(t))^2\mathbb{1}\left[C \leq t\right] + \frac{\mathbb{E}_T\,\mathbb{1}\left[T > t\right]}{\mathbb{E}_{T'}\,\mathbb{1}\left[T' > t\right]}G_\theta(t)^2\mathbb{1}\left[C > t\right]\right]$$

$$= \mathop{\mathbb{E}}_{T,C}\left[\frac{(1-G_\theta(t))^2\mathbb{1}\left[C < T\right]\mathbb{1}\left[C \leq t\right]}{\mathbb{E}_{T'}\,\mathbb{1}\left[C < T'\right]} + \frac{G_\theta(t)^2\mathbb{1}\left[T > t\right]\mathbb{1}\left[C > t\right]}{\mathbb{E}_{T'}\,\mathbb{1}\left[T' > t\right]}\right]$$

$$= \mathop{\mathbb{E}}_{T,C}\left[\frac{(1-G_\theta(t))^2\mathbb{1}\left[C < T\right]\mathbb{1}\left[C \leq t\right]}{P_\theta(T' > C)} + \frac{G_\theta(t)^2\mathbb{1}\left[T > t\right]\mathbb{1}\left[C > t\right]}{P_\theta(T' > t)}\right]$$

$$= \mathop{\mathbb{E}}_{T,C}\left[\frac{(1-G_\theta(t))^2\mathbb{1}\left[C < T\right]\mathbb{1}\left[U \leq t\right]}{P_\theta(T' > U)} + \frac{G_\theta(t)^2\mathbb{1}\left[U > t\right]}{P_\theta(T' > t)}\right]$$

$$= \text{G-BS-CW}(t)$$

The expectation comes out due to $T \perp\!\!\!\perp C$. The last line follows from $C < T \implies U = C$ (in the left term) and $\mathbb{1}[T > t]\,\mathbb{1}[C > t] = \mathbb{1}[U > t]$ (in the right term).

## D  Negative Bernoulli Log Likelihood

Negative BLL is similar to BS, but replaces the squared error with negated log loss:

$$\text{NBLL}(t; \theta) = \underset{T,C,X}{\mathbb{E}} \left[ -\log(F_{\theta_T}(t \mid X))\mathbb{1}[T \le t] - \log(\overline{F}_{\theta_T}(t \mid X))\mathbb{1}[T > t] \right]$$

IPCW BLL can likewise be written as [Kvamme et al., 2019]:

$$\text{F-NBLL-CW}(t; \theta) = \underset{T,C,X}{\mathbb{E}} \left[ \frac{-\log(F_{\theta_T}(t \mid X))\Delta\mathbb{1}[U \le t]}{G(U^- \mid X)} + \frac{-\log(\overline{F}_{\theta_T}(t \mid X))\mathbb{1}[U > t]}{G(t \mid X)} \right]$$

## E  Game Algorithm

---

**Algorithm 2** Following Gradients in Multi-Player Games

---

**Input:** Choice of losses $\ell_F, \ell_G$, learning rate $\gamma$
**Initialize** $\theta_{Tt}$ and $\theta_{Ct}$ randomly for $t = 1, \ldots, K-1$
**repeat**
    **//**  for each parameter of each player
    **for** $t = 1$ **to** $K-1$ **do**
        $g_{Tt} \leftarrow d\ell_F^t/d\theta_{Tt}$
        $g_{Ct} \leftarrow d\ell_G^t/d\theta_{Ct}$
    **end for**
    **//**  for each parameter of each player
    **for** $t = 1$ **to** $K-1$ **do**
        $\theta_{Tt} \leftarrow \theta_{Tt} - \gamma g_{Tt}$
        $\theta_{Ct} \leftarrow \theta_{Ct} - \gamma g_{Ct}$
    **end for**
**until** convergence

---

## F  Experiments

### F.1  Data

**Gamma Simulation**  We draw $x$ from a 32 dimensional multivariate normal $\mathcal{N}(0, 10I)$. We simulate conditionally gamma failure times with mean $\mu_t$ a log-linear function of $x$ with coefficients for each feature drawn $\text{Unif}(0, 0.1)$. The censoring times are also conditionally gamma with mean $0.9 * \mu_t$. Both distributions have constant variance $0.05$. $\alpha, \beta$ parameterization of the gamma is recovered from mean, variance by $\alpha = \mu^2/\sigma^2$ and $\beta = \mu/\sigma^2$. $T$ and $C$ are conditionally independent given $X$. Each random seed draws a new dataset.

We report metrics as a function of training size. We use training sizes [200,400,600,800,1000]. We use validation size 1024 and testing size 2048.

**Survival MNIST**  Survival-MNIST [Gensheimer, 2019, Pölsterl, 2019] draws times conditionally on MNIST label $Y$. This means digits define risk groups and $T \perp\!\!\!\perp X \mid Y$. Times within a digit are i.i.d. The model only sees the image pixels $X$ as covariates so it must learn to classify digits (risk groups) to model times. The PyCox package [Kvamme et al., 2019] uses Exponential times. We follow Goldstein et al. [2020] and use Gamma times. $T$'s mean is $10 * (Y + 1)$ so that lower labels $Y$ mean sooner event times. We set the variance constant to $0.05$. $C$ is drawn similarly but with $9.9 * (Y + 1)$. Each random seed draws a new dataset.

We report metrics as a function of training size. We use training sizes [512, 1024, 2048, 4096, 8192, 10240]. We use validation size 1024 and testing size 2048.

**Real Data** We report results on

- SUPPORT [Knaus et al., 1995] which includes severely ill hospital patients. There are 14 features. we split into 5,323 for training, 1774 for validation, and 1776 for testing.
- METABRIC [Curtis et al., 2012]. There are 9 features. We split into 1,142 for training, 380 for validation, and 382 for testing.
- ROTT [Foekens et al., 2000] and GBSG [Schumacher et al., 1994] combined into one dataset (ROTT. & GBSG). There are 7 features. We split into 1,339 for training, 446 for validation, and 447 for testing.

For more description see Therneau [2021], Katzman et al. [2018], Chen [2020].

In the main text, we report results on a subset of these datasets with metrics as a function of training size. We use training sizes [10, 20, 30, 40, 50, 60, 70, 80, 90, 100, 110, 120, 130, 140, 150, 175, 200]. We use validation size 300 and always use the entire testing set. We standardize all real data with the training set mean and standard deviation.

## F.2 Models

In all experiments except for MNIST, we use a 3-hidden-layer ReLU network. The hidden sizes are [128, 64, 64] for the Gamma simulation and [128,256,64] for the real data. We output 20 categorical bins. See appendix G.1 for different choices of number of bins, which did not show any significant differences in results. For MNIST we first use a small convolutional network and follow with the same fully-connected network, but using hidden sizes [512,256,64].

## F.3 Training

We use learning rate $0.001$ in all experiments for all losses using the Adam optimizer. We train for 300 epochs for the simulated data and 200 for the real data. For all data and all losses, this was enough to overfit on the training data. We use no weight decay or dropout.

## F.4 Model Selection

We select the best model on the validation set using the following approach:

1. Save the $F$ and $G$ models from all the epochs in $F$-set and $G$-set.
2. Randomly choose a model $\tilde{F}$ in the $F$-set.
3. Use $\tilde{F}$ as the weight for $\ell_G$. Find the model $\tilde{G}$ from $G$-set to minimize $\ell_G$ weighted by $\tilde{F}$.
4. Use $\tilde{G}$ as the weight for $\ell_F$. Find the model $\tilde{F}$ from $F$-set to minimize $\ell_F$ weighted by $\tilde{G}$.
5. Repeat steps 3 and 4 until convergence.

Once converged, we use $\tilde{F}$ and $\tilde{G}$ as our best model to evaluate at the test set. The above approach plays as similar role as the game. Instead of gradient descent, this time we select a model from a set to play the game. We first fix $F$ to find the best $G$ based on $\ell_G$ and then fix $G$ to find the best $F$ based on $\ell_F$.

# G Ablations

## G.1 Changing number of bins on MNIST

**Changing number of categorical bins (K) in [10,20,30,40,50]. Cannot directly compare between two choices of K due to changing meaning of likelihood/BS/Concordance but can compare NLL and BS-Game at each K. Trends similar across all choices of K.**

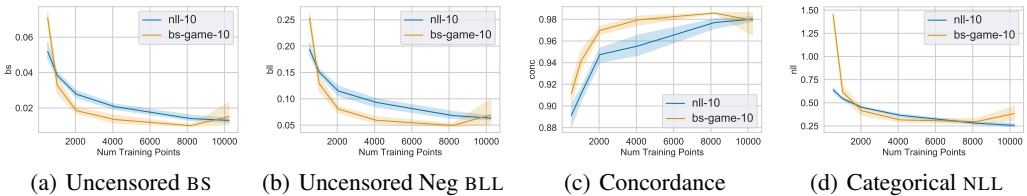

|  (a) Uncensored BS | (b) Uncensored Neg BLL | (c) Concordance | (d) Categorical NLL |

**Figure 9:** 10 bins. NLL (Blue). BS-Game (Orange).

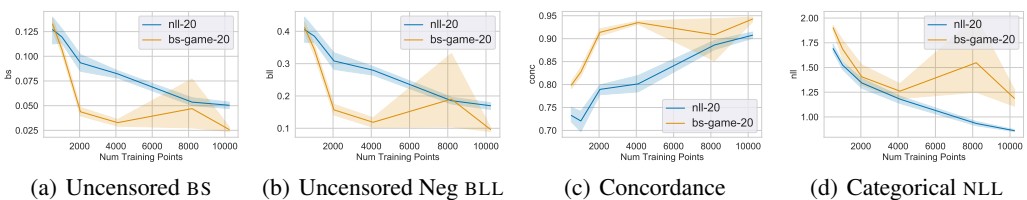

|  (a) Uncensored BS | (b) Uncensored Neg BLL | (c) Concordance | (d) Categorical NLL |

**Figure 10:** 20 bins. NLL (Blue). BS-Game (Orange).

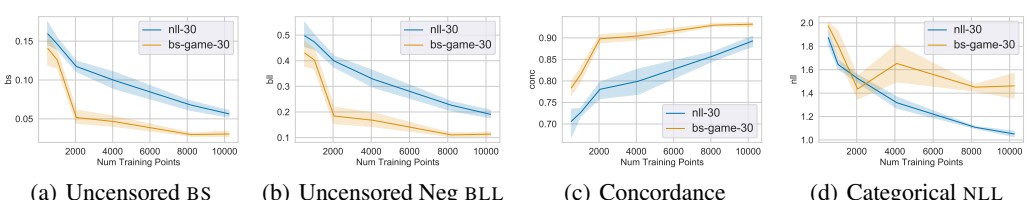

|  (a) Uncensored BS | (b) Uncensored Neg BLL | (c) Concordance | (d) Categorical NLL |

**Figure 11:** 30 bins. NLL (Blue). BS-Game (Orange).

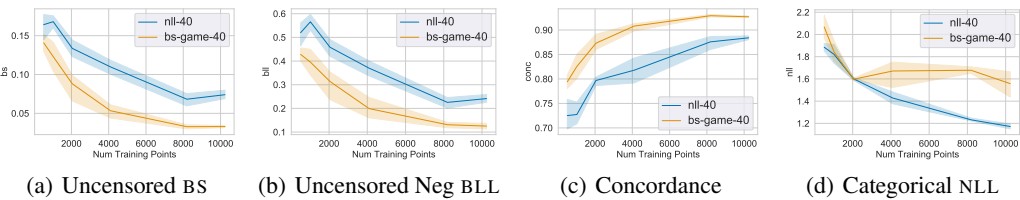

|  (a) Uncensored BS | (b) Uncensored Neg BLL | (c) Concordance | (d) Categorical NLL |

**Figure 12:** 40 bins. NLL (Blue). BS-Game (Orange).

# H Proof of Summed or Integrated Brier Score to be proper

**Proposition 3.** Assume we have a list of time $t_1, \ldots, t_K$. Assume the true distribution for $T$ is $F^* = F_{\theta_T^*}$ in eq. (2). We have:

- The summed BS $\sum_{i=1}^K BS(t_i; \theta)$ is proper, i.e., it has one minimizer at the true parameters $\theta_T^*$.

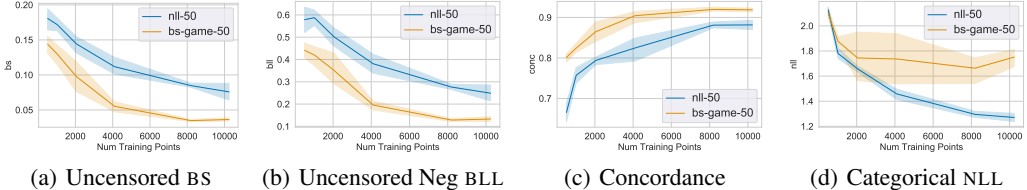

(a) Uncensored BS    (b) Uncensored Neg BLL    (c) Concordance    (d) Categorical NLL

**Figure 13:** 50 bins. NLL (Blue). BS-Game (Orange).

- The integrated BS $\int_{t_1}^{t_K} BS(t; \theta) dt$ is proper, i.e., it has one minimizer at the true parameters $\theta_T^*$.

*Proof.* Since BS$(t)$ is proper, it has one minimizer at $\theta_T^\star$, i.e., for $\theta_T \neq \theta_T^*$, BS$(t; \theta_T^\star) \leq$ BS$(t; \theta_T)$ for all $t$. Since this holds for all $t$, we then have:

$$\sum_{i=1}^{K} BS(t_i; \theta_T^\star) \leq \sum_{i=1}^{K} BS(t_i; \theta_T).$$

This means that the summed Brier Score at $\theta_T^*$ is smaller than at any other $\theta_T$. The summed BS has one minimizer at the true parameters $\theta_T^*$, i.e., it is proper. Since the BS inequality holds for all $t$, we also have

$$\int_{t_1}^{t_K} BS(t; \theta_T^\star) dt \leq \int_{t_1}^{t_K} BS(t; \theta_T) dt$$

This means that the integrated Brier Score at $\theta_T^*$ is smaller than at any other $\theta_T$. The integrated BS has one minimizer at the true parameters $\theta_T^*$, i.e., it is proper. $\qquad\square$

## I   Proof of proposition 1

Here we prove that the true solution is a stationary point of the game. We restate the proposition here.

**Proposition.** Assume $\exists \theta_T^\star \in \Theta_T, \exists \theta_C^\star \in \Theta_C$ such that $F^\star = F_{\theta_T^\star}$ and $G^\star = G_{\theta_C^\star}$. Assume the game losses $\ell_F, \ell_G$ are based on proper losses $L$ and that the games are only computed at times for which positivity holds. Then $(\theta_T^\star, \theta_C^\star)$ is a stationary point of the game eq. (4).

$$\ell_F(\theta) = L_I(F_{\theta_T}; G_{\theta_C}), \quad \ell_G(\theta) = L_I(G_{\theta_C}; F_{\theta_T}) \tag{4}$$

*Proof.* In $\ell_F(\theta)$, by the definition of the IPCW estimator, when $\theta_C = \theta_C^*$, $L_I(F_{\theta_T}; G_{\theta_C}) = L(F_{\theta_T})$. Due to the fact that $L$ is proper, $\theta_T^*$ is a minimizer for $L(F_{\theta_T})$. Then at $(\theta_T, \theta_C) = (\theta_T^*, \theta_C^*)$, we have

$$\left. \frac{d\ell_F(\theta)}{d\theta_T} \right|_{\substack{\theta_T = \theta_T^* \\ \theta_C = \theta_C^*}} = \left. \frac{dL_I(F_{\theta_T}; G_{\theta_C^*})}{d\theta_T} \right|_{\theta_T = \theta_T^*} = \left. \frac{dL(F_{\theta_T})}{d\theta_T} \right|_{\theta_T = \theta_T^*} = 0$$

Similarly for $\ell_G(\theta)$, we have

$$\left. \frac{d\ell_G(\theta)}{d\theta_C} \right|_{\substack{\theta_C = \theta_C^* \\ \theta_T = \theta_T^*}} = \left. \frac{dL_I(G_{\theta_C}; F_{\theta_T^*})}{d\theta_C} \right|_{\theta_C = \theta_C^*} = \left. \frac{dL(G_{\theta_C})}{d\theta_C} \right|_{\theta_C = \theta_C^*} = 0$$

Since the two gradients are zero, the game will stay at the true parameters. Therefore, $(\theta_T^\star, \theta_C^\star)$ is a stationary point of the game eq. (4). $\qquad\square$

## J   Proof of proposition 2

Here we prove that under one construction of the game in algorithm 2, the true solution is the unique stationary point of the game. We restate the proposition here.

**Proposition.** Consider discrete distributions over $K$ times. Let $\theta_T = \{\theta_{T1}, \cdots, \theta_{T(K-1)}\}$, $\theta_{Tt} = P_\theta(T = t)$, $F_{\theta_T}(t) = \sum_{k=1}^{t} \theta_{Tk}$, and likewise for $C, \theta_C$. Assuming that $\theta^\star_{Tt} > 0$ and $\theta^\star_{Ct} > 0$, the solution $(\theta^\star_T, \theta^\star_C)$ is the only stationary point for the multi-player BS game shown in algorithm 2 for times $t \in \{1, \ldots, K-1\}$

*Proof.* We show by induction on the time $t$ of the IPCW BS game that the simultaneous gradient equations are only satisfied at $\hat{\theta}_T = \theta^\star_T$ and $\hat{\theta}_C = \theta^\star_C$. There is a lot of arithmetic but eventually it comes down to (1) substitution of one variable for another (2) assuming all previous timestep parameters are correct (induction) (3) finding the zeros of a quadratic (4) showing that one of the two solutions is the correct parameter and the other is invalid.

**Note:** this proof uses the notation that $\hat{\theta}$ is a model parameter and $\theta^\star$ is the correct one.

### J.1   BS(1) (base case)

We can compute the expectations defining F-BS-CW(1) and G-BS-CW(1) in closed form. That gives us:

$$\text{F-BS-CW}(1) = \theta^\star_{T1}(1 - \hat{\theta}_{T1})^2 + (1 - \theta^\star_{T1})(1 - \theta^\star_{C1})\frac{\hat{\theta}^2_{T1}}{1 - \hat{\theta}_{C1}}$$

$$\text{G-BS-CW}(1) = \frac{\theta^\star_{C1}(1 - \theta^\star_{T1})(1 - \hat{\theta}_{C1})^2}{1 - \hat{\theta}_{T1}} + (1 - \theta^\star_{T1})(1 - \theta^\star_{C1})\frac{\hat{\theta}^2_{C1}}{1 - \hat{\theta}_{T1}}$$

The derivatives are

$$\frac{d\text{F-BS-CW}(1)}{d\hat{\theta}_{T1}} = 2\frac{(1 - \theta^\star_{T1})(1 - \theta^\star_{C1})}{1 - \hat{\theta}_{C1}}\hat{\theta}_{T1} - 2(1 - \hat{\theta}_{T1})\theta^\star_{T1} = 0$$

$$\frac{d\text{G-BS-CW}(1)}{d\hat{\theta}_{C1}} = 2\frac{(1 - \theta^\star_{T1})(1 - \theta^\star_{C1})}{1 - \hat{\theta}_{T1}}\hat{\theta}_{C1} - 2\frac{(1 - \theta^\star_{T1})(1 - \hat{\theta}_{C1})\theta^\star_{C1}}{1 - \hat{\theta}_{T1}} = 0$$

We can take each derivative equation and write one variable in terms of the other. First, taking $d\text{F-BS-CW}/d\hat{\theta}_{T1}$ and writing $\hat{\theta}_{T1}$ in terms of $\hat{\theta}_{C1}$:

$$\frac{d\text{F-BS-CW}(1)}{d\hat{\theta}_{T1}} = 2\frac{(1 - \theta^\star_{T1})(1 - \theta^\star_{C1})}{1 - \hat{\theta}_{C1}}\hat{\theta}_{T1} - 2(1 - \hat{\theta}_{T1})\theta^\star_{T1} = 0$$

implies

$$\frac{(1 - \theta^\star_{T1})(1 - \theta^\star_{C1})}{1 - \hat{\theta}_{C1}}\hat{\theta}_{T1} = (1 - \hat{\theta}_{T1})\theta^\star_{T1}$$

$$\frac{(1 - \theta^\star_{T1})(1 - \theta^\star_{C1})}{1 - \hat{\theta}_{C1}}\hat{\theta}_{T1} + \theta^\star_{T1}\hat{\theta}_{T1} = \theta^\star_{T1}$$

$$\left(\frac{(1 - \theta^\star_{T1})(1 - \theta^\star_{C1})}{1 - \hat{\theta}_{C1}} + \theta^\star_{T1}\right)\hat{\theta}_{T1} = \theta^\star_{T1}$$

$$\hat{\theta}_{T1} = \frac{\theta^\star_{T1}}{\left(\frac{(1-\theta^\star_{T1})(1-\theta^\star_{C1})}{1-\hat{\theta}_{C1}} + \theta^\star_{T1}\right)}$$

Now solving for $\hat{\theta}_{C1}$ in the G-BS-CS derivative:

$$\frac{d\text{G-BS-CW}(1)}{d\hat{\theta}_{C1}} = 2\frac{(1 - \theta^\star_{T1})(1 - \theta^\star_{C1})}{1 - \hat{\theta}_{T1}}\hat{\theta}_{C1} - 2\frac{(1 - \theta^\star_{T1})(1 - \hat{\theta}_{C1})\theta^\star_{C1}}{1 - \hat{\theta}_{T1}} = 0$$

implies

$$\frac{(1 - \theta^\star_{T1})(1 - \theta^\star_{C1})}{1 - \hat{\theta}_{T1}}\hat{\theta}_{C1} = \frac{(1 - \theta^\star_{T1})(1 - \hat{\theta}_{C1})}{1 - \hat{\theta}_{T1}}\theta^\star_{C1}$$

Given $1 - \theta^\star_{T1} \neq 0$ and $1 - \hat{\theta}_{T1} \neq 0$, we have

$$(1 - \theta^\star_{C1})\hat{\theta}_{C1} = (1 - \hat{\theta}_{C1})\theta^\star_{C1}$$

which gives us $\hat{\theta}_{C1} = \theta^\star_{C1}$. Given $1 - \theta^\star_{T1} \neq 0$ and $1 - \hat{\theta}_{T1} \neq 0$, the above derivative equations jointly imply

$$\hat{\theta}_{T1} = \left(\theta^\star_{T1}\right)\left(\frac{(1 - \theta^\star_{T1})(1 - \theta^\star_{C1})}{1 - \hat{\theta}_{C1}} + \theta^\star_{T1}\right)^{-1}, \quad \hat{\theta}_{C1} = \theta^\star_{C1}$$

Substituting $\hat{\theta}_{C1} = \theta^\star_{C1}$ in the formula for $\hat{\theta}_{T1}$ in terms of $\hat{\theta}_{C1}$, we have

$$\hat{\theta}_{T1} = \left(\theta^\star_{T1}\right)\left(\frac{(1 - \theta^\star_{T1})(1 - \theta^\star_{C1})}{1 - \theta^\star_{C1}} + \theta^\star_{T1}\right)^{-1} = \frac{\theta^\star_{T1}}{(1 - \theta^\star_{T1}) + \theta^\star_{T1}} = \theta^\star_{T1}$$

Therefore, under the assumptions, for the BS(1) case, we have the only stationary point at the two true 1st-timestep parameters: $\hat{\theta}_{T1} = \theta^\star_{T1}$ and $\hat{\theta}_{C1} = \theta^\star_{C1}$.

## J.2 Induction step

We can proceed by induction over timesteps. Claim: given $P_\theta(T \leq a) = P^\star(T \leq a)$ and $P_\theta(C \leq a) = P^\star(C \leq a)$, $a = 1, \ldots, k$, the stationary point of the game BS(k+1) has to satisfy $P_\theta(T = k + 1) = P^\star(T = k + 1)$ and $P_\theta(C = k + 1) = P^\star(C = k + 1)$ i.e. $\hat{\theta}_{T,k+1} = \theta^\star_{T,k+1}$ and $\hat{\theta}_{C,k+1} = \theta^\star_{C,k+1}$. We first simplify F-BS-CW.

$$\text{F-BS-CW}(k + 1) = \underset{T,C}{\mathbb{E}}\left[\frac{(1 - F_\theta(k + 1))^2 \mathbb{1}\left[T \leq C\right] \mathbb{1}\left[U \leq k + 1\right]}{P_\theta(C' \geq U)} + \frac{F_\theta(k + 1)^2 \mathbb{1}\left[U > k + 1\right]}{P_\theta(C' > k + 1)}\right]$$

We simplify each term of F-BS-CW separately. The left term of F-BS-CW is

$$\mathop{\mathbb{E}}_{T,C} \frac{(1 - F_\theta(k+1))^2 \mathbb{1}\left[T \le C\right] \mathbb{1}\left[U \le k+1\right]}{P_\theta(C' \ge U)}$$

$$= P_\theta(T > k+1)^2 \mathop{\mathbb{E}}_{T,C} \frac{\mathbb{1}\left[T \le C\right] \mathbb{1}\left[U \le k+1\right]}{P_\theta(C' \ge U)}$$

$$= P_\theta(T > k+1)^2 \sum_{a=1}^{K} \sum_{b=1}^{K} P^\star(T=a) P^\star(C=b) \frac{\mathbb{1}\left[a \le b\right] \mathbb{1}\left[\min(a,b) \le k+1\right]}{P_\theta(C' \ge \min(a,b))}$$

$$\left[\text{condition } \mathbb{1}\left[a \le b\right] \text{ moves from indicator to sum limits and } \min(a,b) = a\right]$$

$$= P_\theta(T > k+1)^2 \sum_{a=1}^{K} \sum_{b=a}^{K} \frac{P^\star(T=a) P^\star(C=b) \mathbb{1}\left[a \le k+1\right]}{P_\theta(C' \ge a)}$$

$$\left[\text{condition } \mathbb{1}\left[a \le k+1\right] \text{ moves from indicator to sum limit}\right]$$

$$= P_\theta(T > k+1)^2 \sum_{a=1}^{k+1} \sum_{b=a}^{K} \frac{P^\star(T=a) P^\star(C=b)}{P_\theta(C' \ge a)}$$

$$= P_\theta(T > k+1)^2 \sum_{a=1}^{k+1} P^\star(T=a) \sum_{b=a}^{K} \frac{P^\star(C=b)}{P_\theta(C' \ge a)}$$

$$= P_\theta(T > k+1)^2 \sum_{a=1}^{k+1} P^\star(T=a) \frac{P^\star(C \ge a)}{P_\theta(C' \ge a)}$$

$$\left[\text{induction hypothesis: } P_\theta(C \le a) = P^\star(C \le a), \quad a = 1, \ldots, k \implies P_\theta(C > a) = P^\star(C > a), \quad a = 1, \ldots, k\right]$$

$$= P_\theta(T > k+1)^2 \sum_{a=1}^{k+1} P^\star(T=a) \cdot 1$$

$$= P_\theta(T > k+1)^2 P^\star(T \le k+1)$$

$$= (1 - \sum_{i=1}^{k} \hat{\theta}_{Ti} - \hat{\theta}_{T(k+1)})^2 \sum_{i=1}^{k+1} \theta^\star_{Ti}$$

$$\left[\text{induction hypothesis: } P_\theta(T \le a) = P^\star(T \le a), \quad a = 1, \ldots, k\right]$$

$$= (1 - \sum_{i=1}^{k} \theta^\star_{Ti} - \hat{\theta}_{T(k+1)})^2 \sum_{i=1}^{k+1} \theta^\star_{Ti}$$

$$= (1 - p - x)^2 (p + t)$$

$$\stackrel{\Delta}{=} A, \qquad \text{where } p = \sum_{i=1}^{k} \theta^\star_{Ti}, q = \sum_{i=1}^{k} \theta^\star_{Ci}, x = \hat{\theta}_{T(k+1)}, y = \hat{\theta}_{C(k+1)}., t = \theta^\star_{T(k+1)} c = \theta^\star_{C(k+1)}.$$

The right term of F-BS-CW is

$$\underset{T,C}{\mathbb{E}} \frac{F_\theta(k+1)^2 \mathbb{1}[U > k+1]}{P_\theta(C' > k+1)}$$

$$= \frac{F_\theta(k+1)^2}{P_\theta(C' > k+1)} \underset{T,C}{\mathbb{E}} \mathbb{1}[U > k+1]$$

$$\left[ T \text{ and } C \text{ are independent means } \mathbb{1}[U > z] = \mathbb{1}[T > z]\, \mathbb{1}[C > z] \right]$$

$$= \frac{F_\theta(k+1)^2}{P_\theta(C' > k+1)} P^\star(T > k+1) P^\star(C > k+1)$$

$$= \frac{(\sum_{i=1}^{k+1} \hat{\theta}_{Ti})^2}{1 - \sum_{i=1}^{k+1} \hat{\theta}_{Ci}} (1 - \sum_{i=1}^{k+1} \theta_{Ti}^\star)(1 - \sum_{i=1}^{k+1} \theta_{Ci}^\star)$$

$$\left[ \text{induction hypothesis: } P_\theta(T \le a) = P^\star(T \le a) \text{ and } P_\theta(C \le a) = P^\star(C \le a), \quad a = 1, \ldots, k \right]$$

$$= \frac{(\sum_{i=1}^{k} \theta_{Ti}^\star + \hat{\theta}_{T(k+1)})^2}{1 - \sum_{i=1}^{k} \theta_{Ci}^\star - \hat{\theta}_{C(k+1)}} (1 - \sum_{i=1}^{k+1} \theta_{Ti}^\star)(1 - \sum_{i=1}^{k+1} \theta_{Ci}^\star)$$

$$= \frac{(p+x)^2}{1-q-y}(1-p-t)(1-q-c) \triangleq B$$

where again $p = \sum_{i=1}^{k} \theta_{Ti}^\star, q = \sum_{i=1}^{k} \theta_{Ci}^\star, x = \hat{\theta}_{T(k+1)}, y = \hat{\theta}_{C(k+1)}, t = \theta_{T(k+1)}^\star, c = \theta_{C(k+1)}^\star$.
To summarize, F-BS-CW$(k+1) = A + B$:

$$\text{F-BS-CW}(k+1) = (1-p-x)^2(p+t) + \frac{(p+x)^2}{1-q-y}(1-p-t)(1-q-c)$$

Then we simplify G-BS-CW.

$$\text{G-BS-CW}(k+1) = \underset{T,C}{\mathbb{E}} \left[ \frac{(1-G_\theta(k+1))^2 \mathbb{1}[C < T]\, \mathbb{1}[U \le k+1]}{P_\theta(T' > U)} + \frac{G_\theta(k+1)^2 \mathbb{1}[U > k+1]}{P_\theta(T' > k+1)} \right]$$

The left term of G-BS-CW

$$\mathop{\mathbb{E}}_{T,C} \frac{(1 - G_\theta(k+1))^2 \mathbb{1}\left[C < T\right] \mathbb{1}\left[U \le k+1\right]}{P_\theta(T' > U)}$$

$$=(1 - G_\theta(k+1))^2 \mathop{\mathbb{E}}_{T,C} \frac{\mathbb{1}\left[C < T\right] \mathbb{1}\left[U \le k+1\right]}{P_\theta(T' > U)}$$

$$=(1 - G_\theta(k+1))^2 \sum_{a=1}^{K}\sum_{b=1}^{K} P^\star(C = a)P^\star(T = b)\frac{\mathbb{1}\left[a < b\right]\mathbb{1}\left[\min(a,b) \le k+1\right]}{P_\theta(T' > \min(a,b))}$$

condition $\mathbb{1}\left[a < b\right]$ moves from indicator to sum limits and $\min(a,b) = a$

$$=(1 - G_\theta(k+1))^2 \sum_{a=1}^{K}\sum_{b=a+1}^{K} \frac{P^\star(C = a)P^\star(T = b)\mathbb{1}\left[a \le k+1\right]}{P_\theta(T' > a)}$$

condition $\mathbb{1}\left[a \le k+1\right]$ moves from indicator to sum limits

$$=(1 - G_\theta(k+1))^2 \sum_{a=1}^{k+1}\sum_{b=a+1}^{K} \frac{P^\star(C = a)P^\star(T = b)}{P_\theta(T' > a)}$$

$$\left[\text{split sum over a into two terms: 1 through k, and k+1, recall b starts at a+1}\right]$$

$$=(1 - G_\theta(k+1))^2 \left(\sum_{a=1}^{k}\sum_{b=a+1}^{K} \frac{P^\star(C = a)P^\star(T = b)}{P_\theta(T' > a)} + \sum_{b=k+2}^{K} \frac{P^\star(C = k+1)P^\star(T = b)}{P_\theta(T' > k+1)}\right)$$

$$=(1 - G_\theta(k+1))^2 \left(\sum_{a=1}^{k} P^\star(C = a)\sum_{b=a+1}^{K} \frac{P^\star(T = b)}{P_\theta(T' > a)} + P^\star(C = k+1)\sum_{b=k+2}^{K} \frac{P^\star(T = b)}{P_\theta(T' > k+1)}\right)$$

$$=(1 - G_\theta(k+1))^2 \left(\sum_{a=1}^{k} \frac{P^\star(C = a)P^\star(T \ge a+1)}{P_\theta(T' > a)} + \frac{P^\star(C = k+1)P^\star(T > k+1)}{P_\theta(T' > k+1)}\right)$$

$$=(1 - G_\theta(k+1))^2 \left(\sum_{a=1}^{k} \frac{P^\star(C = a)P^\star(T > a)}{P_\theta(T' > a)} + \frac{P^\star(C = k+1)P^\star(T > k+1)}{P_\theta(T' > k+1)}\right)$$

$$\left[\text{induction hypothesis: } P_\theta(T \le a) = P^\star(T \le a), \quad a = 1,\ldots,k \implies P_\theta(T > a) = P^\star(T > a), \quad a = 1,\ldots,k\right]$$

$$=(1 - G_\theta(k+1))^2 \left(\sum_{a=1}^{k} P^\star(C = a) + \frac{P^\star(C = k+1)P^\star(T > k+1)}{P_\theta(T' > k+1)}\right)$$

$$=(1 - \sum_{i=1}^{k}\hat{\theta}_{Ci} - \hat{\theta}_{C(k+1)})^2 \left(\sum_{i=1}^{k} \theta^\star_{Ci} + \frac{\theta^\star_{C(k+1)}(1 - \theta^\star_{T(k+1)} - \sum_{i=1}^{k}\theta^\star_{Ti})}{1 - \sum_{i=1}^{k}\hat{\theta}_{Ti} - \hat{\theta}_{T(k+1)}}\right)$$

$$\left[\text{induction hypothesis: } P_\theta(T \le a) = P^\star(T \le a) \quad \text{and} \quad P_\theta(C \le a) = P^\star(C \le a), \quad a = 1,\ldots,k\right]$$

$$=(1 - \sum_{i=1}^{k}\theta^\star_{Ci} - \hat{\theta}_{C(k+1)})^2 \left(\sum_{i=1}^{k} \theta^\star_{Ci} + \frac{\theta^\star_{C(k+1)}(1 - \theta^\star_{T(k+1)} - \sum_{i=1}^{k}\theta^\star_{Ti})}{1 - \sum_{i=1}^{k}\theta^\star_{Ti} - \hat{\theta}_{T(k+1)}}\right)$$

$$=(1 - q - y)^2(q + \frac{c(1 - t - p)}{1 - p - x}) \triangleq C$$

By symmetry with $B$, the right term is

$$\mathop{\mathbb{E}}_{T,C} \frac{G_\theta(k+1)^2 \mathbb{1}\left[U > k+1\right]}{P_\theta(T' > k+1)} = \frac{(q + y)^2}{1 - p - x}(1 - q - c)(1 - p - t) \triangleq D$$

Again using $p = \sum_{i=1}^{k} \theta_{Ti}^{\star}, q = \sum_{i=1}^{k} \theta_{Ci}^{\star}, x = \hat{\theta}_{T(k+1)}, y = \hat{\theta}_{C(k+1)}, t = \theta_{T(k+1)}^{\star}, c = \theta_{C(k+1)}^{\star}$, we have

$$\text{G-BS-CW}(k+1) = C + D$$

$$= (1 - q - y)^2 (q + \frac{c(1 - t - p)}{1 - p - x}) + \frac{(q + y)^2}{1 - p - x}(1 - q - c)(1 - p - t)$$

The stationary point satisfies

$$\frac{\partial \text{G-wt-FBS(k+1)}}{\partial x} = \frac{\partial A}{\partial x} + \frac{\partial B}{\partial x}$$

$$= -2(1 - p - x)(p + t) + 2\frac{(p + x)}{1 - q - y}(1 - p - t)(1 - q - c)$$

$$= 0$$

$$\frac{\partial \text{F-wt-GBS(k+1)}}{\partial y} = \frac{\partial C}{\partial y} + \frac{\partial D}{\partial y}$$

$$= -2(1 - q - y)(q + \frac{c(1 - t - p)}{1 - p - x}) + 2\frac{(q + y)}{1 - p - x}(1 - q - c)(1 - p - t) = 0$$

It's a system of quadratic equations with two unknowns. The system has analytical solutions. Solving the above equations for $x, y$ by *Mathematica* (it is quite a long derivation manually), the solutions are

$$x = t, y = c$$

or

$$x = (1/(-q + q^2 + qc))(cp - qcp - qt + q^2 t + ct$$
$$- (p(-1 + q + c + qp - q^2 p - cp + qt - q^2 t - ct))/((-1 + q)(p + t))$$
$$+ (qp(-1 + q + c + qp - q^2 p - cp + qt - q^2 t - ct))/((-1 + q)(p + t))$$
$$- (t(-1 + q + c + qp - q^2 p - cp + qt - q^2 t - ct))/((-1 + q)(p + t))$$
$$+ (qt(-1 + q + c + qp - q^2 p - cp + qt - q^2 t - ct))/((-1 + q)(p + t)))$$
$$y = (-1 + q + c + qp - q^2 p - cp + qt - q^2 t - ct)/((-1 + q)(p + t))$$

To check if this second solution is valid, it would need to be the case that $q + y < 1$ because we only consider $k + 1 < K$. If we ask mathematica to simplify q+y that satisfies the above solution, then this holds:

$$q + y = \frac{-1 + q - c(-1 + p + t)}{(-1 + q)(p + t)}$$

The numerator and the denominator are both negative. If $k + 1 < K$ (we know BS at K is 0 and also we only have K-1 parameters), the numerator minus denominator =

$$-1 + q - c(-1 + p + t) - (-1 + q)(p + t) = (-1 + q)(1 - p - t) - c(-1 + p + t)$$
$$= (-1 + q + c)(1 - p - t)$$
$$< 0$$

Therefore,

$$\sum_{i=1}^{k} \theta_{Ci}^{\star} + \hat{\theta}_{C(k+1)} = q + y > 1$$

This is invalid. So

$$x = t, y = c$$

is the only solution, i.e., $\hat{\theta}_{T(k+1)} = \theta_{T(k+1)}^{\star}, \hat{\theta}_{C(k+1)} = \theta_{C(k+1)}^{\star}$. By induction, we conclude that

$$\hat{\theta}_{Ti} = \theta_{Ti}^{\star}, \hat{\theta}_{Ci} = \theta_{Ci}^{\star}, i = 1, \ldots, K - 1$$

By $\hat{\theta}_{TK} = 1 - \sum_{i=1}^{K-1} \hat{\theta}_{Ti}$ and $\hat{\theta}_{CK} = 1 - \sum_{i=1}^{K-1} \hat{\theta}_{Ci}$, we have

$$\hat{\theta}_{TK} = \theta_{TK}^{\star}, \hat{\theta}_{CK} = \theta_{CK}^{\star}$$

Therefore,

$$\hat{\theta}_{Ti} = \theta_{Ti}^{\star}, \hat{\theta}_{Ci} = \theta_{Ci}^{\star}, i = 1, \ldots, K$$

is the only stationary point for the game. □