# OpenReview forum: "Inverse-Weighted Survival Games"
_NeurIPS.cc/2021/Conference — NeurIPS 2021 Poster_

### Official Review · Reviewer_MkMh · 2021-07-16

**Rating:** 6
**Confidence:** 3

**Summary:**

The paper introduces a non-adversarial approach for simultaneously learning events and censoring cumulative distribution functions (CDF) at prespecified discrete-time points by optimizing inverse weighted Brier score (BS) and Bernoulli log-likelihood (BLL). Further, the theoretical analysis illustrates that the proposed approach achieves data event and censoring distributions at stationary points. Experimental results on (semi-)synthetic and real-world datasets demonstrate that the proposed achieves better concordance, Brier score, and Bernoulli log-likelihood than an approach that optimizes the standard negative log-likelihood (NLL) given small sample sizes.

**Limitations And Societal Impact:**

The discussion needs to include the limitations of discrete-time models and the chosen metrics (BS, BLL).

**Main Review:**

Overall, the paper is well organized and easy to follow.  Moreover, provided theoretical analysis supports the recovery of the data distribution at stationary points. However, below are the key limitations.

**Significance**
- The paper focuses on discrete-time models, which are challenging to scale for large time horizons. Can the proposed approach be adapted to extensive continuous-time survival methods? Also, the paper title needs to be updated to include discrete-time survival since this is the focus of the submission.
- The paper needs to motivate the utility of the proposed metrics (time-varying BS, BLL) relative to standard metrics such as concordance index, relative absolute error, etc. Also, why are integrated BS(BLL) not considered?
- While the BS (or BLL) are proper scoring rules, is there evidence to support that IPCW-BS(BLL) are also proper scoring rules?

**Underwhelming experimental results**
- The paper should also benchmark against other approaches besides negative log-likelihood (NLL), including CoxPH and alternative approaches that optimize BS, such as [1].
- Experimental results on real-world datasets demonstrate that the proposed approach has no clear advantage over the NLL.
- Also, why are the sample sizes on real-world datasets limited to 200? The SUPPORT dataset has ~10,000 observations.
- The paper should also provide qualitative results,  e.g.,  predicted CDFs against ground truth events or censoring times.

**Below are the key missing references**
- [1]  Avati et al., "Countdown regression: sharp and calibrated survival predictions", In Uncertainty in Artificial Intelligence, 2020.

--------------------------------------------------------------------------------------------------------------------------------------------------------------------------
I thank the authors for addressing my concerns and providing additional experimental results. Hence, I have increased my score.



**Time Spent Reviewing:**

5 hours

---

> ### Author Response · Authors · 2021-08-11
> **Response to Reviewer MkMh**
>
> **[Overall, the paper is well organized and easy to follow]**
>
> Thanks!
>
> **[Can the proposed approach be adapted to extensive continuous-time?]**
>
> Our prop 1 that the true distribution is always a stationary point of IPCW games holds true of continuous distributions as well. We have edited the text to emphasize this. For prop 2 (uniqueness of stationary point for BS) our proof technique was specific to discrete distributions and extending it remains for future work.
>
> **[paper title needs to be updated to include discrete-time]**
>
> Gladly. Perhaps “Deep Discrete Survival Games” or “Inverse Weighted Games for Discrete-Time Survival Analysis”.
>
> **[motivate utility of proposed metrics (time-varying BS, BLL) relative to standard metrics]**
>
> We would like to note that we do not propose these metrics but build on recent literature that uses them. First, we provide additional motivation. Then, we provide additional references.
>
> Beyond the properties of being proper + having the right stationary points, we can further motivate BS/BLL(t) from the perspective of calibration.
>
> In survival analysis, calibration is valued since often models are used to communicate risk. Most calibration errors measure absolute or squared loss between modeled CDF values and empirical CDFs. The minimizer of BS(t)/BLL(t) is P(T>t|X), which fits the conditional CDF and therefore measures conditional calibration. Additionally, BS can be decomposed into MSE and marginal calibration error  [DeGroot + Fienberg. The comparison and evaluation of forecasters. 1983].
>
> To motivate IPCW BLL, we note that one difficulty in interpreting the NLL is that its scale is censoring distribution dependent.
>
> BS/BLL are reported in recent machine learning for survival analysis papers in JMLR, Stats in Medicine, AISTATS, ICML. For example, for BS:
> - Haider et al. Effective Ways to Build and Evaluate Individual Survival Distributions JMLR 2020  *The Brier score is a commonly used metric that measures both calibration and discrimination* (which is referring to DeGroot’s decomposition of BS into MSE + (Marginal) Calibration error)
> - Steingrimsson and Morrison. Deep learning for survival outcomes. Statistics in medicine. 2020
> - Lee et al. Temporal Quilting. AISTATS 2019: *As the metric of calibration, we use the BS which is the mean square error adjusted for the survival setting*
> - Kumar et al. Trainable Calibration Measures For Neural Networks From Kernel Mean Embeddings. ICML 2018. Uses BS as evaluation for calibration.
> - Kvamme et al.. Time to Event Prediction with NNs and Cox. JMLR 2019. *The BS for binary classification is a metric of both discrimination and calibration of a model’s estimates*
> - R package “pec” for evaluating survival models and python libraries Sci-kit Survival and PySurvival.
>
> **[Also, why are integrated BS(BLL) not considered?]**
>
> We hope to clarify: the integrated (summed) BS/BLL are considered in this work: each model optimizes the time-varying BS/BLL at each timestep: good BS at all time steps implies good summed BS.
>
> Additionally, here we share results for optimizing the summed game rather than “for all timesteps” game. All trends are similar to the original submission results but game methods improved.
> https://drive.google.com/file/d/1GsurmR89z6uMVY3WMR4YcnHSRcQpQq5i/view?usp=sharing
>
> **[While the BS (or BLL) are proper scoring rules, is there evidence to support that IPCW-BS(BLL) are also proper scoring rules?]**
>
> “Proper” is traditionally defined for objectives of a single distribution. IPCW games depend on 2 distributions. We therefore need to establish the right meaning of “proper” for two models.
>
> First, If G is fixed at the truth, then F’s objective is proper since it equals true (under no censoring) BS/BLL, which is proper.
>
> Next, our Figure 1 shows that if you optimize the sum of F and G’s IPCW BS/BLL, the resulting objective is not proper for the pair (F,G).
>
> Our main contribution is game training. Here “proper” (that true distributions are minimums) is replaced by stationary points (places where the game stops).
>
> The question we address: “is the pair of true F,G a stationary point of the game?”. We conclude that our theorems and experimental evidence provide a positive answer to a rephrasing of the reviewer’s question which pertains to games featuring two distributions rather than objectives for one distribution.
>
> **[paper should benchmark against CoxPH and Countdown Regression’s S-CRPS objective]**
>
> Optimizing the Cox partial likelihood does not optimize BS for general models/distributions. S-CRPS on the other hand, under appropriate assumptions, may be able to:
> - S-CRPS is proper under censoring (as is NLL)
> - S-CRPS also coincides with an estimator of the unweighted IBS, meaning that it is a biased estimator of the true BS under censoring.
> - In more detail, from IPCW, we know that there is some covariate-dependent weight required in the estimator when censoring is conditional. Even when censoring is marginal there is some covariate-independent but censoring-distribution-specific and bin-specific constant for each time bin required to estimate true BS in each bin. We often don’t know if the true censoring distribution is marginal or conditional.
> - Advantage of S-CRPS: could do well at optimizing true BS in finite samples if the missing constants due to marginal censoring are not too different in each bin which implies that censoring is near marginally uniform. S-CRPS has a great advantage of not requiring access to a censoring distribution estimate.
> - Caveats of S-CRPS: when censoring is conditional, or non-uniform marginal, S-CRPS should not be expected to directly optimize true BS in finite samples (though the infinite data limit’s solution coincides with the solution to BS).
> - Take-away: one should validate (using whichever criteria are available) among (1) methods that are safe because they can account for conditional censoring (e.g. games) and (2) methods that are safe because they do not require censoring estimates (e.g. NLL and S-CRPS)
>
> We ran a discrete summed version of S-CRPS on the METABRIC dataset and find (1) S-CRPS does not directly optimize the BS and BLL as the games do (expected) and (2) S-CRPS is competitive on concordance https://drive.google.com/file/d/1PdcTVbgXNhx-wlpaFwAH-MPxoH2KboYb/view?usp=sharing We are currently adding this method to all experiments.
>
> **[Experimental results on real-world datasets demonstrate that the proposed approach has no clear advantage over the NLL.]**
>
> We would like to clarify the meaning of the plots in case we have miscommunicated them.
>
> We believe the conclusion “no clear advantage” may have resulted from reading the plots as training curves where the result is the right-hand end of the plot. This is not what the plots report.
>
> Instead, for each data size separately (looking at a vertical slice at a time), this plot shows test-set performance of the best model from a converged training procedure. “Best” is chosen on a validation set. This is measured across several seeds.
>
> For example, judging by the plots for METABRIC (Figure 5) We see that at separate experiments for each training size, the game methods achieve better test set BS than the NLL-training method. We likewise see this for concordance.
>
> **[Also, why are the sample sizes on real-world datasets limited to 200? The SUPPORT dataset has ~10,000 observations]**
>
> In the text we mention “these survival datasets are low-dimensional, so any of the objectives can perform well on the metrics with just several hundred datapoints”.  Test-set performance curves converge around 250 training points. We validate and test on the rest of the available dataset as mentioned in the appendix.
>
> Looking at the SUPPORT plots (Figure 7) we see that BS, BLL, and NLL have converged (with respect to training data size) for all 3 training methods.
>
> **[Paper should provide qualitative results, e.g., predicted CDFs against ground truth events or censoring times]**
>
> This is a good suggestion that coincides with another review.
>
> We first clarify: The BS/BLL objectives do compare model CDFs against ground truth events.
>
> Most calibration errors measure absolute or squared loss between modeled CDF values and empirical CDFs. The minimizer of BS(t)/BLL(t) is P(T>t|X), which fits the conditional CDF and therefore measures conditional calibration. Additionally, BS can be decomposed into MSE and marginal calibration error  [DeGroot + Fienberg. The comparison and evaluation of forecasters. 1983] where marginal calibration also assesses the model CDF.
>
> We would be glad to include additional metrics that assess predicted CDFs or other qualitative measures. Could the reviewer please provide specific references to definitions/examples/estimators of  such additional metrics? Thanks!
>
> **[missing references. Countdown Regression]**
>
> We have added discussion of S-CRPS into the text
>
> **[discussion needs to include the limitations of discrete-time and chosen metrics]**
>
> - Limitations for discrete time models: bins may be wrong in relevant time ranges
> - Limitations for weighted BS/BLL: weights may be wrong.

---

> > ### Comment · Reviewer_MkMh · 2021-09-14
> > **Response to Rebuttal**
> >
> > I thank the authors for addressing most of my concerns.  Also, I appreciate the effort it took to provide additional experimental results.
> >
> > **Experimental Results**
> >
> > Overall, while the theoretical results are interesting, the experimental results are underwhelming. For instance, I don't think the main paper should include convergence curves which may confuse the reader.  Also, though Cox-PH does not optimize BS/BLL, it is widely used in practice; hence benchmarking with Cox-PH is crucial.
> >
> > **Qualitative Calibration Plots**
> >
> > Refer to [1, 2] for examples of qualitative population comparisons of calibration results.
> >
> > [1]  Avati et al., "Countdown regression: sharp and calibrated survival predictions", In Uncertainty in Artificial Intelligence, 2020.
> >
> > [2] Chapfuwa et al., "Calibration and Uncertainty in Neural Time-to-Event Modeling", In IEEE-TNNLS, 2020.

---

> > > ### Author Response · Authors · 2021-09-16
> > > **Clarifications, Qualitative Results, Additional Experiments**
> > >
> > > Thank you for the response and further comments.
> > >
> > > **[experimental results + clarifying "convergence curve"]**
> > >
> > > We would like to clarify: our plots **do not** show each method's performance as a function of training epochs. In figures 3,4,5,6,7, every point along each curve is the test set performance of a model **fully-trained** on a training set with size equal to the X axis tick.  These figures show the improvements that game-training offers over NLL-training for small training data sizes.
> > >
> > > We have updated our figure captions to clarify this.
> > >
> > > We would be happy to address any remaining concerns about the experiments, if the reviewer can let us know precisely which results are underwhelming and why.
> > >
> > > **[Cox PH]**
> > >
> > > As suggested, we fit a CoxPH model on the gamma simulation data.
> > >
> > > The game-trained models outperform CoxPH on test BLL and test concordance metrics at all training set sizes. On test BS metric, CoxPH and game-trained models are similar at training set size 200, but game-trained models are better for all larger sizes.
> > >
> > > **[qualitative calibration plots]**
> > >
> > > As suggested, we have included a qualitative calibration plot comparing game-training and NLL-training.
> > >
> > > https://drive.google.com/file/d/1SHlOLAQyIrUkK96rIs3fbUhCY225YVUV/view?usp=sharing.
> > >
> > > In this plot, BS-game achieves near perfect calibration while the two likelihood-based methods suffer some calibration error. This ties back to [our earlier discussion](https://openreview.net/forum?id=j4oYd8SGop&noteId=wdtvNpXX8LA) on motivating BS from a calibration perspective.
> > >
> > > ---
> > >
> > > Your comments have helped us improve the quality of the paper. If, as you mentioned, we have addressed most of your concerns, we would appreciate it if you could update your rating to reflect this.
> > >
> > > Thanks again,
> > > Authors

---

### Official Review · Reviewer_mDpZ · 2021-07-16

**Rating:** 7
**Confidence:** 4

**Summary:**

In this paper the authors exploit the symmetry present in the survival analysis setting between the roles of C and Y to frame the problem as a game with the parameters as (unique with certain conditions) solution.
The authors prove then provide experiments showing not only that the approach is competitive but can even outperform the standard approach in the low data regime.

**Limitations And Societal Impact:**

The authors adequately addressed the limitations and potential negative societal impact of their work?

**Main Review:**

The authors adapt the IPCW framework, which is known to have good generalization properties (see [1]) even when the weights have to be estimated with a twist: instead of estimating S_C first once and for all, they iteratively refine the estimations of S_C and S_Y be exploiting the fact they both symmetrically are the solution to an IPCW ERM problem.


[1]G. Ausset, S. Clémençon, and F. Portier, “Empirical Risk Minimization under Random Censorship: Theory and Practice,”. Available: http://arxiv.org/abs/1906.01908

The main contribution is the proof that for certain losses the solution of interest is indeed a stationary point and that for a particular loss, it is the only saddle point.

The arguments are presented clearly and are convincing but some questions may benefit from more details:

1. The approach proposed is conceptually very similar to the doubly robust approach [2], has the link been studied by the authors?

[2] D. Rubin and M. J. van der Laan, “A Doubly Robust Censoring Unbiased Transformation,” The International Journal of Biostatistics, vol. 3, no. 1, Jan. 2007, doi: 10.2202/1557-4679.1052.

2. More details on the training procedure would be welcome (I personally never managed to achieve convergence using this method.)

Finally, some typos in the manuscript remain, some of them unfortunate (line 244: not invariant => invariant).

Overall I think this is a very interesting work and a worthwhile contribution.

**Time Spent Reviewing:**

4

---

> ### Author Response · Authors · 2021-08-11
> **Response to Reviewer mDpZ**
>
> **[G. Ausset, S. Clémençon, and F. Portier, “Empirical Risk Minimization under Random Censorship: Theory and Practice,”]**
>
> Thanks for the reference! We have incorporated this highly related work into our writing.
>
> **[arguments are presented clearly and are convincing]**
>
> Thanks!
>
> **[conceptually similar to Double Robust Censoring Unbiased Transformations]**
>
> Great question on the related doubly robust (DR) literature! In that text, they DR-estimate e.g. $\mathbb{E}[T]$ or $\mathbb{E}[T|X]$ where the estimator works with a correct failure model $F_\theta$ or a correct censoring model $G_\theta$.
>
> Let $h(\theta,x,t)$ compute the BS loss of $F_\theta$ on a single datapoint $(x,t)$. Then, in our text, we would like to estimate $\mathbb{E}[h(\theta,T,X)]$ where $h$ necessarily must compute $F_\theta(T|X)$ CDF values. In addition to the censoring model a DR estimator for this quantity would require a model that computes $h(\theta,T,X)$'s expectation given $X$ under (X,T) samples from the true distribution (itself a survival problem).
>
> Since our purpose for estimating $\mathbb{E}[h]$ is to estimate an objective for training $F_\theta$, it seems like it would pre-suppose a solution to the failure modeling problem to require estimation/access to a model $x \rightarrow h$. We however note that once someone believes $F_\theta$ and $G_\theta$ approximate $F,G$ well, you are correct that you could apply this DR estimator to evaluate the BS! We have added this discussion to our text and will continue to consider this interesting direction.
>
> **[2. More details on the training procedure would be welcome]**
>
> We have updated our appendix to include more training details and are committed to releasing easy-to-use code with the complete paper.
>
> **[typos , line 244: not invariant => invariant]**
>
> Interestingly, we *did* mean “not invariant” rather than “invariant”. It turns out the common “only comparable pairs” concordance also needs re-weighting. Otherwise you are only reporting concordance on a population whose failure times tend to occur before other people’s failure or censoring times, which is a subpopulation. Interesting concordance is also improper (Blanche et al. The C-index is not proper for evaluation of t-year predicted risks).
>
> **[Overall I think this is a very interesting work and a worthwhile contribution]**
>
> Thank you for your close read and the relevant literature suggestions!

---

### Official Review · Reviewer_XC9X · 2021-07-17

**Rating:** 6
**Confidence:** 5

**Summary:**

This paper proposes a new training method for discrete-time survival models, where not only a failure model but also a censoring model are simultaneously trained to optimize a set of time-dependent inverse probability of censor-weighting (IPCW) criteria. Discrete-time models trained with different objective functions have been compared on simulated data and multiple real datasets.

**Limitations And Societal Impact:**

Limitations have been discussed. More potential limitation: the game method is currently developed for the discrete-time models. But the inverse probability game seems to be general enough for continuous-time survival models as well, which may mitigate the issue of a large number of simultaneous objectives in the IPCW game for discrete-time models.

**Main Review:**

Strength:
- The new training method that directly optimizes IPCW criteria for discrete-time models is novel and has been shown to have better test BS, BLL, and concordance on simulated data, especially for small sample sizes.
- They show that the IPCW game method has a stationary point at the data-generating distributions and IPCW BS has only one stationary point.
- The paper is generally well written and easy to follow.

Weakness:
- This paper claims that "deep discrete-time models have become a default choice even when data is continuous", this is not true as there are some recent deep continuous-time survival models [1,2] that are shown to have advantages over discrete-time models.
- The ultimate goal is to obtain an estimated event time distribution that is close to the truth. Different evaluation criteria characterize how far the estimated distribution is from the truth in different perspectives, while the common optimum is expected to be the truth. Why optimizing the BS and BLL is particularly preferable / desired in the context of survival? I find the current argument "a conceptually simple way to evaluate models that is particularly suitable for discrete models is to apply binary classification losses at various time horizons, for instance Brier score (BS) for 5-year survival, or similarly, Bernoulli log-likelihood" is weak. The authors do find some empirical improvement on other criteria when using the new training method (BS, BLL based), but high-level intuition on why it is expected to work better would be appreciated.
- How is 20 selected as the number of categorical bins in experiments? The number of objectives to simultaneously optimize in the proposed method scales linearly with the number of bins, but, in general, discrete-time survival models require enough bins to well approximate a continuous event time. This would affect both the computation complexity and the numerical stability. Additional experiments would be desired.
- The assumption used in building the IPCW objective ($T \perp C| X$) and that used in evaluation metrics ($T \perp C$) for real datasets are inconsistent.
- I am trying to understand why the proposed method works better for small sample sizes. This observation is relatively counterintuitive for me. Because the game method simultaneously models the failure time and censoring time distribution and thereby contains more parameters than the NLL method. But it works better for small sample sizes.
- References need to be updated such as [1] is still cited as its arXiv version. Please check other citations as well.

References:

[1] H. Kvamme, Ørnulf. Borgan, and I. Scheel. Time-to-event prediction with neural networks and cox regression. Journal of Machine Learning Research, 20(129):1–30, 2019.
[2] Tang, W., J. Ma, Q. Mei, and J. Zhu (2020). SODEN: A scalable continuous-time survival model through ordinary differential equation networks.arXiv preprint arXiv:2008.08637.


-------------------------------------------------------
I thank the authors for their thorough responses. After the discussion, I raised my evaluation.

**Time Spent Reviewing:**

6

---

> ### Author Response · Authors · 2021-08-11
> **Response to Reviewer XC9X**
>
> **[Strength: new training method, novel, shown to have better test BS, BLL, and concordance, well written and easy to follow]**
>
> Thanks!
>
> **[Claim that deep discrete models are default is not true, exist good recent deep continuous model results]**
>
> Our apologies, we agree the word “default” was too strong. We did not intend to suggest that continuous time models are never more appropriate. Instead, we emphasize at most that deep discrete models have become popular, as seen in recent papers in JMLR, NeurIPS, AISTATS etc:
> - Yu et al. MTLR: Learning Patient-Specific Cancer Survival Distributions as a Sequence of Dependent Regressor NeurIPS 2011.
> - Lee et al. Deephit. AAAI 2018
> - Miscouridou et al. Deep Survival Analysis: Nonparametrics and Missingness. MLHC 2018.
> - Lee et al. Temporal Quilting. AISTATS 2019.
> - Ren et al. Deep Recurrent Survival Analysis. AAAI 2019
> - Kvamme Borgan and Scheel. Time to Event Prediction with NNs and Cox. JMLR 2019.
> - Goldstein et al. Explicit Calibration for Survival Analysis. NeurIPS 2020.
> - Haider et al. Effective ways to build and evaluate individual survival distributions. JMLR 2020.
> - Sloma et al. Empirical Comparison of Continuous and Discrete-time Representations for Survival Prediction. AAAI Symposium on Survival Prediction 2021.
> - Kamran and Wiens. Estimating Calibrated Individualized Survival Curves with Deep Learning. AAAI 2021
>
>
> **Theoretical results and Discrete/Continuous**: our prop 1 that the true distribution is always a stationary point of IPCW games holds true of continuous distributions as well. We have edited the text to emphasize this. For prop 2 (uniqueness of stationary point for BS) our proof technique was specific to discrete distributions and extending it remains for future work.
>
> **[Motivation on optimization of BS/BLL. Why preferable? Why does it work?]**
>
> We should remove the weak phrasing “conceptually simple”, agreed. Beyond the properties of being proper + having the right stationary points, we can further motivate BS/BLL(t) from the perspective of calibration.
>
> BS/BLL(t)’s minimizer for every t is $P(T>t|X)$. In survival analysis, calibration is valued since often models are not used just to make point predictions but to communicate risk at time horizons. Several common definitions of calibration compute squared error of model CDFs and observed incidence rates. BS fits this CDF error with squared loss.
>
> Next, to motivate IPCW BLL, we note that one difficulty in interpreting the NLL is that its scale is censoring distribution dependent. On the other hand, the BLL still measures log likelihood but is reweighed to be invariant to the censoring distribution.
>
> We also remark that BS/BLL are reported in recent machine learning for survival analysis papers in JMLR, Stats in Medicine, AISTATS, ICML. For example, for BS:
> - Haider et al. Effective Ways to Build and Evaluate Individual Survival Distributions JMLR 2020. *The Brier score is a commonly used metric that measures both calibration and discrimination* (which is referring to DeGroot’s decomposition of BS into MSE + (Marginal) Calibration error)
> - Steingrimsson and Morrison. Deep learning for survival outcomes. Statistics in medicine. 2020
> - Lee et al. Temporal Quilting. AISTATS 2019: *As the metric of calibration, we use the BS which is the mean square error adjusted for the survival setting*
> - Kumar et al. Trainable Calibration Measures For Neural Networks From Kernel Mean Embeddings. ICML 2018. Uses BS as evaluation for calibration.
> - Kvamme et al.. Time to Event Prediction with NNs and Cox. JMLR 2019. *The BS for binary classification is a metric of both discrimination and calibration of a model’s estimates.*
> - R package “pec” for evaluating survival models and python libraries Sci-kit Survival and PySurvival.
>
>
> **[stability of multiple objectives + selection of 20 cat bins]**
>
> **numerical stability**: instead of the “for all time steps” games we find that optimizing summed F-IPCW-BS(t) across time with respect to all F parameters, and likewise for G, is also stable. It can ease stability since this means the model parameters are following the gradient field of a scalar function.
>
> In theory, summing removes the uniqueness of the infinite-data IPCW BS game stationary point but in practice it finds good solutions (and there are always additional stationary points due to deep conditional parameterizations anyway).
>
> Here are the results with the summed game. All trends are similar to the original submission results but game methods improved.
> https://drive.google.com/file/d/1GsurmR89z6uMVY3WMR4YcnHSRcQpQq5i/view?usp=sharing
>
> **For complexity**: doesn’t seem to be much that can be done unless the weighted BS has a closed form for some continuous distribution.
>
> **Regarding # bins**: we ran MNIST experiments comparing NLL-training and Game-training at # bins in [10,20,30,40,50] and see similar trends to the standard 20 bins.
> https://drive.google.com/file/d/1PKQldImiQ4w9U_rHh9jq-31KpQeQl0hK/view?usp=sharing
>
> **[Training assumption T indep C given X and real data eval assumption T indep C are inconsistent]**
>
> - They do not strictly contradict each other: T|X1 and C|X2 for independent X1,X2 satisfies both.
> - Suppose that T indep C (evaluation assumption holds). Does T indep C given X (training assumption) necessarily hold? Yes, except for when X is a common child (collider) of T,C, but this does not happen in survival analysis due to time ordering (event times don’t generate baseline covariates)
> - Additionally, the training is also okay under marginally independent censoring T indep C. We have updated the text to mention this.
> - Finally, what if the training assumption T indep C given X holds but not the evaluation assumption T indep C marginally?
>
> As we mention in experiments + conclusion, there is never ground truth for censoring with real data, so evaluation is impossible without assumptions. Following the precedent in a few published papers like the JMLR paper [1] we made this assumption (T indep C) to make evaluation possible and because survival studies such as METABRIC/SUPPORT may plausibly have marginal censoring.
>
> [1] Kvamme et al. Time-to-event prediction with neural networks and cox regression. JMLR 2019.
>
> **[counterintuitive: game contains more parameters than the NLL method. But games works better for small sample sizes]**
>
> It’s a great question. In general, choosing an objective is stating a preference for how finite sample errors are allocated.
>
> First, why would NLL not directly optimize BS?
> - NLL and BS/BLL have the same optima in the limit of samples: a model that has perfectly solved the NLL objective on infinite samples will have optimal BS due to NLL/BS being proper.
> - But one should not expect NLL to directly optimize BS since estimates of NLL are not estimates of BS (neither unbiased nor consistent).
>
> Next, discussion on more vs less parameters:
>
> - The game approach has no more parameters than optimizing NLL separately for the failure and for the censoring distribution.
> - NLL for the failure and censoring distribution separately has more parameters than just doing NLL for the failure distribution, but the failure distribution performs no worse when you optimize censoring than when you don’t because censoring is assumed to be independent and is modeled independently.
> - The game likewise models two independent distributions.
> - The additional challenge in the game over separated NLL is the coupling of the errors: one wrong model affects the other model’s objective.
> - We therefore have a tradeoff, but not exactly a bias/variance tradeoff: NLL’s two models don’t affect each others’ losses while the game has coupled objectives but can target BS or BLL directly, which NLL can never do (see first set of bullet points): at near-convergence of the G model, in the game the F model optimizes near-unbiased estimates of the evaluation criteria directly.
>
>
> We would like practitioners to add this approach to their toolbox --not replace NLL-- when they desire models with good BS/BLL.
>
> **[Kvamme citation should be updated from arXiv to JMLR]**
>
> Thanks for catching this! We have updated our text.
>
> **[More potential limitation: discrete time. Continuous may hold too]**
>
> As mentioned, our prop 1 holds for continuous time and we are interested in exploring the continuous setting in follow up work
>
> **Summary:**
> Thank you for the tremendously helpful and specific feedback. We hope that our additional discussion and references for BS and discrete models clarifies the motivation/premise of the paper.
>
> If the additional reasoning about evaluation assumptions and “more/less parameters” provides insight , and if the additional results on number of bins and summed objective help round out the experiments, then we would be grateful if you consider updating your rating of this work.

---

> > ### Comment · Reviewer_XC9X · 2021-09-13
> > **Additional comments on the response**
> >
> > I thank the authors for seriously addressing my comments and I still have some remaining comments.
> >
> > **[Focus on discrete-time model]**: Referring to continuous-time methods [1,2,3] as well and a clear summary on the cases in which discrete-time methods are preferred (why they have become popular rather than just listing references) at the beginning would be useful to better position the contribution of this work in the literature. Because the main contribution, IPCW BS/BLL game, is designed for discrete-time models in terms of constructing the loss, algorithm, etc.
> >
> > **[Motivation on optimization of BS/BLL]**: I agree that calibration is important and that's also a part of the reason that BS/BLL has been widely used as the evaluation metric in the literature. Adding your response to the manuscript would be useful. But I don't think the scale of NLL is censoring distribution dependent. Under uninformative censoring, the likelihood is given by $\prod_{i=1}^n p(U_i)^{\Delta_i=1}\bar{F}(U_i)^{\Delta_i=0}$, where $p$ is the pdf of the event time, which does not depend on the censoring distribution.
> >
> > **[stability of multiple objectives + selection of 20 cat bins]**: Thanks for running additional experiments. Adding the experiments on the summed loss and clearly stating the applicability of the theory in the text would be helpful.
> >
> >
> >
> > References:
> >
> > [1] H. Kvamme, Ørnulf. Borgan, and I. Scheel. Time-to-event prediction with neural networks and cox regression. Journal of Machine Learning Research, 20(129):1–30, 2019.
> >
> > [2] Tang, W., J. Ma, Q. Mei, and J. Zhu (2020). SODEN: A scalable continuous-time survival model through ordinary differential equation networks.arXiv preprint arXiv:2008.08637.
> >
> > [3] Steingrimsson, J. A., & Morrison, S. (2020). Deep learning for survival outcomes. Statistics in medicine, 39(17), 2339-2349.

---

> > > ### Author Response · Authors · 2021-09-14
> > > **Updated Text + Clarification about NLL dependence on censoring**
> > >
> > > **[I thank the authors for seriously addressing my comments]**
> > >
> > > Thank you. Incorporating your feedback has improved the quality of the paper.
> > >
> > > **[Focus on discrete-time model]**
> > >
> > > We agree and have updated the background section to motivate discrete models from first principles. We summarize the main points:
> > >
> > > - the integrated/summed BS is analytically tractable for discrete models, so wanting to optimize for BS is one case where a practitioner may directly prefer a discrete model.
> > > - the effects of mis-specification diminish for discrete models as the amount of data and number of bins grow, while a mis-specified continuous model stays biased in the limit. If the practitioner knows enough about the shape of the true failure time distribution, a continuous model may be appropriate. If not, a discrete model is a safer choice.
> > >
> > > We will incorporate the extra citations into the discussion.
> > >
> > > Lastly, we have made explicit in the text that our proposition 1 also holds for continuous distributions.
> > >
> > > **[Motivation on optimization of BS/BLL]**
> > >
> > > Thanks for reading through our argument RE calibration and BS/BLL.
> > >
> > > On the second point, we would like to offer a clarification. We distinguish between the censoring model and the true sampling distribution of censoring times.
> > >
> > > You are correct that the failure model's NLL does not depend on the censoring *model* under independent and non-informative censoring.
> > > However, it does depend on the *true sampling distribution of censoring times*.
> > >
> > > To make this precise, we define the objective using your notation, but making explicit the true sampling distribution.
> > >
> > > $E_{T \sim F_{true}, C \sim G_{true}} [p_\theta(U)^{\Delta=1}  \overline{F}_\theta(U)^{\Delta=0} ]$
> > >
> > > Here, $\Delta$ and $U$ depend on T and C (therefore on $F_{true}$ and $G_{true}$). We now constructively show that the failure model’s NLL can vary with the true censoring distribution. Let us consider a marginal survival analysis problem (no features) and random censoring. The log NLL is:
> > >
> > > $ E_{F_{true},G_{true}}[ \Delta \log p_\theta(U)] + E_{Ftrue,Gtrue}[(1-\Delta)  \log \overline{F}_\theta(U)]$
> > >
> > >
> > > Now consider an $F_{true}$ whose support starts at time $1$ (e.g. uniform over 1,2,3) and $G_{true}$ such that there is probability $\rho$ that $C=0$ and probability $1-\rho$ that $C$ take a value above the support of $T$ (e.g. >3). Points are therefore only censored at time 0 or uncensored.
> > >
> > > $E_{F_{true},G_{true}}[ \Delta \log p_\theta(U)] + E_{F_{true},G_{true}}[(1-\Delta)  \log \overline{F}_\theta(U)]$
> > >
> > > $= (1-\rho) E_{F_{true}}[\log p_\theta(T)] + \rho E_{G_{true}}[\log \overline{F}_\theta(C)]$
> > >
> > > $= (1-\rho)  E_{F_{true}}[\log p_\theta(T)] + \rho E_{G_{true}}[\log \overline{F}_\theta(0)]$
> > >
> > > $= (1-\rho) E_{F_{true}}[\log p_\theta(T)] + \rho E_{G_{true}}[\log 1]$
> > >
> > > $= (1-\rho) E_{F_{true}}[\log p_\theta(T)] + \rho E_{G_{true}}[0]$
> > >
> > > $= (1-\rho) E_{F_{true}}[\log p_\theta(T)]$
> > >
> > > This quantity depends on $\rho$. This shows that the failure model's NLL depends on the true sampling distribution of censoring times.
> > >
> > > **[Adding the experiments on the summed loss and clearly stating the applicability of the theory in the text would be helpful]**
> > >
> > > Thank you. We have included the summed objective results in the text and have added discussion about the theory surrounding the summed and non-summed cases.
> > >
> > > Thank you again for the thorough reviewing.  If we have addressed your concerns, could you update your rating to reflect this? If not, could you let us know any remaining concerns?
> > >
> > > Authors

---

### Official Review · Reviewer_Xu5C · 2021-07-20

**Rating:** 5
**Confidence:** 3

**Summary:**

The authors are interested in the setting of predicting discrete outcomes within time intervals of interest (ex. 5-year survival, 10-year survival) from time-to-event data, i.e. the setting in which some observations may be right-censored. The conventional approach relies on inverse probability of censorship weighting (IPCW), which first estimates the distribution over time-to-censorship $p(c|x)$ (for example, via a Kaplan-Meier estimate), then re-weights the data to estimate the distribution over time-to-event p(t|x)$.

The authors instead propose *Inverse-Weighted Survival Games*, in which they learn *both* the time-to-censorship and the time-to-event distributions in a non-adversarial game, using each to re-weight the other's loss. On the theoretical side, they show that when the loss function is proper, the resulting game has the true distributions as a stationary point. Moreover, when the loss function is the Brier Score, this is the only stationary point. On the experimental side, they run experiments on simulations and public datasets (Survival-MNIST, METABRIC, SUPPORT) and show that IPCW games improve upon NLL-trained models in terms of concordance, KM-weighted log-likelihood, and KM-weighted Brier Score in the data-limited regime.

**Ethical Concerns:**

No ethical concerns.

**Limitations And Societal Impact:**

Yes.

**Main Review:**

I'm a fan of this paper. The idea behind Inverse-Weighted Survival Games is simple and elegant. The background and methods sections are very well-written.

My primary point of concern is with respect to the experiments. I have several questions:

1/ What is the definition of the "NLL"-optimized categorical model? My understanding is that it's a simple optimization of the discrete-time classification model, without accounting for censorship. If so, then without accounting for censorship the metric isn't very useful -- I don't think it'd be worth including in the results.

2/ I'd strongly suggest the authors add as a benchmark a discrete-time categorical model optimized with IPCW via a simple Kaplan-Meier estimate of the censorship distribution $p(c)$. This is important because it's the most common way such models are developed today.

3/ How are the confidence intervals shown in Figures 3-7 derived?

4/ What's the motivation for presenting the plots as a function of the number of training points? I think the training curves will not be relevant to most readers of the paper -- instead, what they'll care about is end-to-end performance. Instead of Figures 3-7, can we have a table of results measuring BS, BLL, Concordance and Calibration at *convergence*? I'd insist on showing confidence intervals for the above as well, for example by showing the standard deviation of each metric over random samples of the evaluation procedure.

5/ I'd strongly sugest the authors include a measure of calibration in their evaluation criteria, alongside concordance (which measures discrimination). One example would be L1 calibration error, where censorship is taken into account via a Kaplan-Meier estimate for the observed incidence rate in each bin.

I'd be happy to revise my rating of this paper if the authors are able to address the above questions.

Miscellaneous typos and comments:

1/ L166: missing commas on $[128, 256, 64$]

2/ How many data points are in the simulation dataset?

3/ What is the rate of censorship in each of the datasets?

**Time Spent Reviewing:**

4

---

> ### Author Response · Authors · 2021-08-11
> **Response to Reviewer Xu5C**
>
> **[I’m a fan of this paper. Idea is simple and elegant. Background and methods are very well-written]**
>
> Thanks for the detailed comments! We appreciate your interest in the paper.
>
> **[1. Definition of "NLL"-optimized model and accounting for censoring]**
>
> Good question. NLL training does make use of censoring times using the standard likelihood approach:
> $\Delta \log p(T|X) + (1-\Delta)\log \big(1-\text{CDF}(C|X)\big)$ which can be found in our appendix A.3 and in papers e.g. [“S_mle-right” on p.3 of https://arxiv.org/pdf/1806.08324.pdf] and [eq. 2 in https://arxiv.org/pdf/1907.00825.pdf] This approach is consistent under random censoring and i.i.d. data, does not rely on modeling the censoring distribution, but the objective value is censoring distribution dependent.
>
> **[2. benchmark via Kaplan-Meier (KM) estimate of the censoring distribution]**
>
> We agree about adding KM-weighted training as an additional benchmark. We first briefly explain what one could expect and then provide a preliminary experiment:
> - Making the marginal assumption (using KM) when censoring is conditional should not yield a performant model under the BS criteria since the training objective does not directly estimate true BS.
> - If in a special case, this incorrectly-weighted BS loss is still proper, then given enough data, the objective yields the true distribution, which has good true BS. But "enough" data is hard to define and depends on the conditional censoring distribution.
> - When marginal censoring does hold, KM estimation is simpler, and could be more or less stable than conditional modeling depending on sample size, data variance, parameterization assumptions. But we may not know if censoring is marginal and therefore assume conditional models.
>
> Here we compare KM-weighting vs full conditional game on MNIST where some digits are heavily censored and others are heavily observed (censoring is conditional).
> https://drive.google.com/file/d/1s_ebm0FAKBr60soyenUj4t36FavpXJgv/view?usp=sharing
> We see that the true BS improves more quickly using the conditional game (blue) than with KM (orange). The test NLL of the KM-weighting method is also poor.
>
> **[3. confidence intervals in plots]**
>
> We show min(lower bound), mean(middle line), and max(upper bound) across 3-5 seeds per experiment.
>
> **[4. motivation of plots as function of datapoints, why isn’t end-to-end performance shown?]**
>
> Let us clarify what seems to be a miscommunication about the plots:
>
> **What the plot is:** For each data size separately, this plot shows test-set performance of the best model from a converged training procedure. “Best” is chosen on a validation set. That is, the plots *do* show end-to-end converged performance with confidence intervals (over random seeds of full training+eval at each size) as requested by the reviewer. The plots show that the IPCW games have better test-set performance on all metrics for small training size, and that all three methods converge on similar performance when there is enough data (all methods prefer the true distribution in expectation)
>
> **What the plot is not:** performance during a single experiment while adding more and more data, That is, this is not a training curve. We have updated the wording to make this distinction very clear.
>
> **[5. Include a measure of calibration e.g. L1]**
>
> Good suggestion to add more calibration evaluation. We first clarify that BS/BLL do measure (certain definitions of) calibration. Next, could we have clarification on the suggested L1 calibration metric to include?
>
> Most calibration errors measure absolute or squared loss between modeled CDF values and empirical CDFs. The minimizer of BS(t)/BLL(t) is P(T>t|X), which fits the conditional CDF and therefore measures conditional calibration. Additionally, BS can be decomposed into MSE and marginal calibration error  [DeGroot + Fienberg. The comparison and evaluation of forecasters. 1983].
>
> For the L1 metric, you refer to absolute error between the modeled CDF and the KM estimate. Is it the D’Agostino-Nam statistic but with absolute error? Could you please provide a reference to a particular equation/estimator/algorithm? We would be glad to include it.
>
> **[How many data points are in the simulation dataset?]**
>
> - Gamma simulation: train with each of [200,400,600,800,1000] points, validate on 1024, and test on 2048.
> - Survival MNIST: train with each of [512,1024,2048,4096,8192,10240].
>
> As mentioned, each training size is a separate experiment run to convergence.
>
> **[What is the rate of censorship in each of the datasets?]**
>
> - Gamma: about 50%,
> - Survival MNIST: about 50%,
> - Rotterdam/GBSG: about 40%,
> - METABRIC: about 40%,
> - SUPPORT: about 30%.
>
> **[happy to revise my rating if authors able to address the questions]**
>
> Beyond some clarification on the additional calibration evaluations to add, we believe all comments have been addressed by the additional results and by adding citations / relevant clarifications.
>
> We would be grateful if you consider a revision to the score given these clarifications, especially the major clarifications about NLL training and meaning of the experiment plots. Thanks!

---

> ### Author Response · Authors · 2021-09-02
> **Score Revision**
>
> Dear Reviewer Xu5C,
>
> Thanks again for your constructive feedback!
>
> You mentioned that you would consider revising your score.
>
> If we have answered your questions and provided clarification, could you raise your score?
>
> If not, could you let us know about any remaining concerns?
>
> Thanks!
> Authors

---

> ### Author Response · Authors · 2021-09-17
> **Following Up**
>
> Dear Reviewer Xu5C,
>
> We would like to confirm whether we have answered your questions.
>
> For reference, here are the main points:
> - the NLL baseline *does* account for censorship
> - we have added the Kaplan-Meier experiment as suggested (a good addition to the paper)
> -  the plots already *do show end-to-end test performance* for several seeds and training set sizes (as opposed to showing convergence during a single training run) as requested in the review. Each point on each curve is a separate complete experiment. We have updated the captions in the text to communicate this.
>
>
> Please let us know if you have any remaining questions or concerns, we would be happy to discuss.
>
> Thanks again,
> Authors

---

### Author Response · Authors · 2021-08-11
**General Response for Inverse Weighted Survival Games**

We thank the Reviewers and the Area Chair for their comprehensive read and critique of our work on inverse-weighted games, a non-adversarial game training method for survival analysis that uses censoring models to help train failure models. Here we summarize the take-aways from the reviews and our responses:

**Common Positives**
We are glad the reviewers found that the presented method is “simple and elegant” (Xu5C), “novel” (XC9X), and “shown to have better test BS, BLL, and concordance” (XC9X). The reviewers describe that “Background and methods are very well-written”(Xu5C),  the writing is “easy to follow” (XC9X, MkMh), and “arguments are presented clearly and are convincing” (mDpZ). Reviewer mDpZ concludes “Overall I think this is a very interesting work and a worthwhile contribution” .

**Common Negatives**
The reviewers requested more motivation for BS/BLL metrics and for the use of discrete models. To this end, we have provided additional reasoning about BS/BLL’s connection to calibration and their invariance to the censoring distribution when reweighted. We have emphasized citations to recent papers in JMLR, NeurIPS, AISTATS, etc. that discuss and report these metrics. Most appear in our submission and we have provided some new ones. We have similarly provided citations to recent publications in major conferences/journals that use discrete models. As mentioned in the submission, we are primarily motivated by discrete models because they can approximate arbitrary continuous distributions as the number of bins increases (no need to specify Gamma, LogNormal, Weibull, etc). This flexibility however comes with higher computational costs. We have also clarified that our proposition 1 holds for continuous models as well.

**Major Clarifications**
The plots may have been understood to show training curves (where the right-hand end of the plot would show the result and one would conclude that all methods perform similarly). Instead, they really show test performance for converged models as a function of each separate model’s training set size. Each vertical slice of the plots shows a comparison of the 3 methods for a given data size. We therefore see in simulations and real datasets that the game approaches have improved test metrics over NLL-training in many instances.

**Additional Experiments Suggested By Reviewers**
1. Reviewer Xu5C asked to try the KM estimator (a non-parametric marginal estimator) for the censoring distribution instead of a conditional model since it’s a simple benchmark. In this experiment, we see that on MNIST with conditional censoring, the conditional model has improved performance over the marginal estimate. https://drive.google.com/file/d/1s_ebm0FAKBr60soyenUj4t36FavpXJgv/view?usp=sharing

2. Reviewer XC9X asked for possible solutions to the issue that the number of objectives scales linearly with the number of categorical bins. To this end, we report results on all datasets for a simpler summed game as opposed to the submitted “for all timesteps” game. Though the summed approach loses the uniqueness of stationary points, in practice  all trends are similar to the original submission results but game methods are improved. https://drive.google.com/file/d/1GsurmR89z6uMVY3WMR4YcnHSRcQpQq5i/view?usp=sharing

3. Reviewer XC9X asked about the choice of 20 categorical bins. Here we ran experiments on survival MNIST for the choices 10,20,30,40,50 and found the results to have similar trends to the original choice of 20 bins. https://drive.google.com/file/d/1PKQldImiQ4w9U_rHh9jq-31KpQeQl0hK/view?usp=sharing

4. Reviewer MkMh asked to implement S-CRPS right-censored objective from the paper Countdown Regression. This objective is similar to an integrated BS (discrete sum in our case) where the portion of the integral after observed censoring times is omitted. We characterize this method as “an unweighted BS that is still proper”. We ran this on the METABRIC dataset and find (1) S-CRPS does not directly optimize the BS and BLL as the games do (expected theoretically, see our response to MkMh) (2) S-CRPS is competitive on concordance. https://drive.google.com/file/d/1PdcTVbgXNhx-wlpaFwAH-MPxoH2KboYb/view?usp=sharing. We are currently adding this method to all experiments.

---

### Decision · Program_Chairs · 2021-09-27

**Decision:**

Accept (Poster)

**Comment:**

To be updated - This is one, where I think the score underrates the work and should probably be accepted once reviewers update their scores based on the author response. I just emailed all of the reviewers who have yet to indicate they have read the author response.